# Microglia remodel synapses by presynaptic trogocytosis and spine head filopodia induction

Laetitia Weinhard [1], Giulia di Bartolomei[1], Giulia Bolasco[1], Pedro Machado[2], Nicole L. Schieber[3], Urte Neniskyte[1,4], Melanie Exiga[1], Auguste Vadisiute[1,4], Angelo Raggioli[1], Andreas Schertel[5], Yannick Schwab[2,3] & Cornelius T. Gross[1]

Microglia are highly motile glial cells that are proposed to mediate synaptic pruning during neuronal circuit formation. Disruption of signaling between microglia and neurons leads to an excess of immature synaptic connections, thought to be the result of impaired phagocytosis of synapses by microglia. However, until now the direct phagocytosis of synapses by microglia has not been reported and fundamental questions remain about the precise synaptic structures and phagocytic mechanisms involved. Here we used light sheet fluorescence microscopy to follow microglia–synapse interactions in developing organotypic hippocampal cultures, complemented by a 3D ultrastructural characterization using correlative light and electron microscopy (CLEM). Our findings define a set of dynamic microglia–synapse interactions, including the selective partial phagocytosis, or trogocytosis (*trogo-*: nibble), of presynaptic structures and the induction of postsynaptic spine head filopodia by microglia. These findings allow us to propose a mechanism for the facilitatory role of microglia in synaptic circuit remodeling and maturation.

[1] Epigenetics and Neurobiology Unit, European Molecular Biology Laboratory (EMBL), Via Ramarini 32, 00015 Monterotondo, Italy. [2] Electron Microscopy Core Facility, European Molecular Biology Laboratory (EMBL), Meyerhofstrasse 1, 69117 Heidelberg, Germany. [3] Cell Biology and Biophysics Unit, European Molecular Biology Laboratory (EMBL), Meyerhofstrasse 1, 69117 Heidelberg, Germany. [4] Department of Neurobiology and Biophysics, Life Science Center, Vilnius University, Sauletekio al. 7, Vilnius 10257, Lithuania. [5] Carl Zeiss Microscopy GmbH, ZEISS Group, Carl-Zeiss-Strasse 22, 73447 Oberkochen, Germany. Correspondence and requests for materials should be addressed to C.T.G. (email: gross@embl.it)

Microglia are glial cells that derive from the myeloid hematopoietic lineage and take up long-lived residence in the developing brain. In response to brain injury, microglia migrate to the site of damage and participate in the phagocytic removal of cellular debris[1]. The recent discovery that microglia are also highly motile in the uninjured brain[2,3], continuously extending and retracting processes through the extracellular space, suggests that they may monitor and contribute to synaptic maturation and function. Clues to this activity come from observations that during early postnatal development microglia undergo morphological maturation that matches synaptic maturation[4], and that they express receptors for neuronal signaling factors that are upregulated during this period[5]. These data, combined with the known phagocytic capacity of myeloid cells, led to the hypothesis that microglia may have a role in the phagocytic elimination of synapses as part of the widespread pruning of exuberant synaptic connections during development[6,7]. This hypothesis was supported by two studies that reported the selective engulfment of synaptic structures by microglia and the appearance of excess immature synapses in mice lacking either the fractalkine[8] (Cx3cl1/Cx3cr1) or complement component[9] (C1q/C3/CR3) microglia signaling pathways.

Numerous studies have confirmed an important role for microglia in promoting synapse and circuit maturation. Knockout (KO) mice lacking complement factors show cortical excitatory hyperconnectivity[10], supporting a role for microglia in the elimination of excess synapses in the mammalian neocortex. KO mice lacking fractalkine receptor show transient excitatory hyperconnectivity followed by weak synaptic multiplicity and reduced functional brain connectivity in adulthood[11,12], suggesting that a failure to eliminate synapses at the correct developmental time prevents the normal strengthening of synaptic connections. Notably, inhibitory synapses in hippocampus appear to be unaffected by disruptions of neuron–microglia signaling[11] (but see ref. [13]). Microglia are also likely to be required for environment-induced brain plasticity as mice lacking microglia P2Y$_{12}$ receptors show deficits in early monocular deprivation-associated visual cortical plasticity[14]. At the same time, studies have pointed to a role for microglia in synapse formation in the adult brain, showing that they can elicit calcium transients and the formation of filopodia from dendritic branches[15], and that they are required for learning-induced synapse formation[16]. Together, these studies suggest that microglia have a complex role in shaping maturing circuits.

An important question raised by these studies is whether synaptic phagocytosis by microglia underlies some or all of these phenotypes. Unfortunately, support for a role of microglia in the phagocytic elimination of synapses is based entirely on indirect evidence—localization of synaptic material within microglia in fixed specimens and increased synapse density following the disruption of microglia function. In this study, we set out to test the hypothesis that microglia engulf and eliminate synapses during mouse hippocampal development. First, we carried out quantitative confocal microscopy analysis of microglia–synapse interactions in fixed hippocampal tissue. Microglia were found to contact dendritic spines, but contrary to previous reports we found no evidence for the elimination of postsynaptic material. Second, we confirmed these data at the ultrastructural level by correlative light and electron microscopy (CLEM) using focused ion beam scanning electron microscopy (FIB-SEM) and discovered evidence for the partial elimination, or trogocytosis, of presynaptic boutons and axons by microglia. Trogocytosis has been described in the immune system as a non-apoptotic mechanism for the rapid capture of membrane components and differs from phagocytosis, which involves the engulfment and elimination of larger cellular structures (> 1 μm)[17–19]. Third, we carried out time-lapse light sheet microscopy of microglia–synapse interactions in organotypic hippocampal cultures and observed the trogocytosis of exclusively presynaptic material. Time-lapse microscopy also revealed the frequent induction of spine head filopodia at sites of microglia–synapse contacts and these were confirmed at the ultrastructural level. Our findings provide the first direct evidence for the elimination of synaptic material by microglia in living brain tissue and suggest that microglia facilitate circuit maturation by a combination of trogocytosis of the axonal compartment and the remodeling of postsynaptic sites.

## Results

**No evidence for phagocytosis of spines by microglia.** To identify the period of maximal microglia phagocytic activity in postnatal hippocampal development, we performed immuno-colocalization analysis of Iba1-labelled microglia with CD68, a phagosomal marker, in fixed sections at postnatal day 8 (P8), P15, P28, and P40 (Supplementary Fig. 1a, b). CD68 immunoreactivity was high at P8–P15 with a peak at P15 and a gradual decrease in the following weeks. The CD68 immunoreactivity pattern at P15 was confirmed in genetically labeled microglia (Supplementary Fig. 1c, d). These findings indicate that the second postnatal week of mouse hippocampal development is likely to be a period of active microglia phagocytosis and suggested that this period may be most relevant to search for evidence of phagocytic elimination of synapses by microglia.

Previous studies have presented indirect evidence for the phagocytic engulfment of dendritic spines in hippocampus and visual cortex[8,14,20]. For example, immunoreactivity for postsynaptic density 95 (PSD95) protein was found to be localized inside microglia by confocal, super-resolution, and electron microscopy[8]. We attempted to confirm these findings using cytoplasmic neuronal and microglia markers, in triple transgenic mice expressing green fluorescent protein (GFP) in sparse excitatory neurons (Thy1::EGFP[21]) and tdTomato in microglia (Cx3cr1::CreER[16]; RC::LSL-tdTomato[22]). We analyzed over 8900 spines from secondary dendrites of GFP + neurons in the CA1 stratum radiatum of fixed hippocampal tissue at P15 and found that about 3% of spines were contacted by microglia (n = 294). The majority of these contacts presented a relatively minor colocalization of spine and microglia fluorescence (apposition, Fig. 1a, c; n = 171, 1.9% of spines), whereas others were characterized by more extensive colocalization as defined by > 70% of the spine surface contacted by microglia (encapsulation, Fig. 1b, c; n = 123, 1.4% of spines). For each encapsulation event we carefully examined the integrity of the spine head, neck, and dendritic shaft, and found that in all cases, even in those where the entire spine surface appeared to be contacted by microglia, an intact GFP + spine neck remained visible (Fig. 1b–d). Thus, using cytoplasmic markers to visualize neuronal and microglia material, we were not able to confirm earlier evidence suggesting the phagocytic engulfment of dendritic spines. To explore whether phagocytic activity might nevertheless be associated with microglia–spine contacts, we performed immunofluorescence colocalization of the phagosomal marker CD68 and microglia–spine contacts. Approximately 15% of microglia–spine contacts (apposition or encapsulation) showed apposed CD68 immunoreactivity (Fig. 1d), suggesting an involvement of local phagocytic activity in the microglia–neuron contact event.

To test this hypothesis we developed a CLEM approach based on previously published methods[24,2523], to first identify rare microglia–spine contact events by fluorescence microscopy and then reconstruct in three dimensions the surrounding ultrastructure by electron microscopy (Fig. 2a and Supplementary Movie 1). Following fixation of hippocampal tissue in a manner compatible with electron microscopy, interaction events were

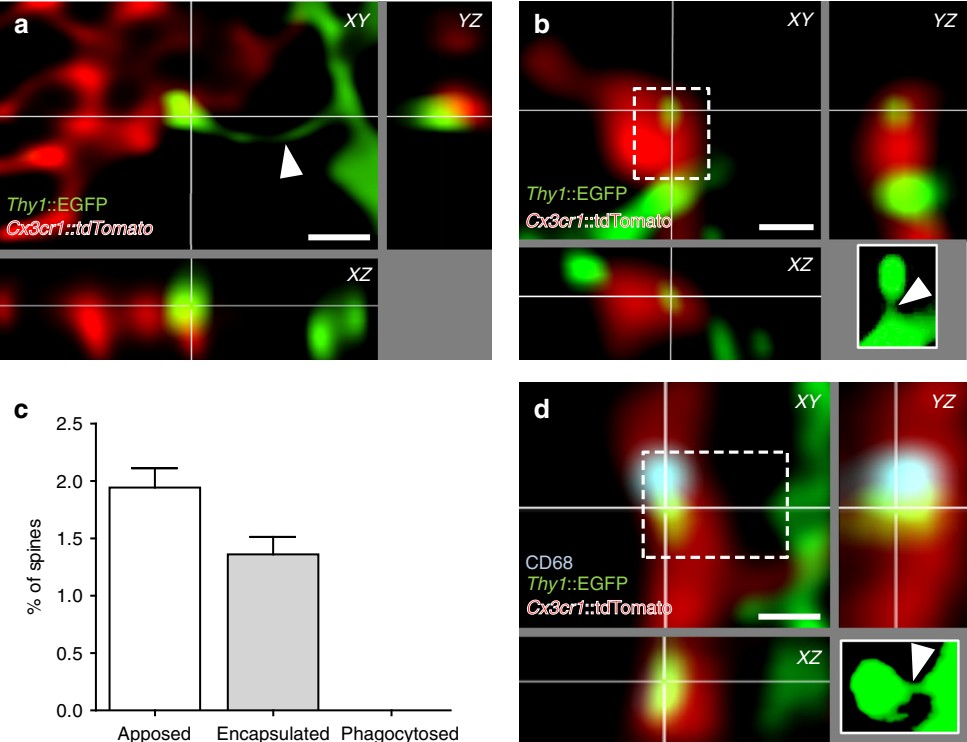

**Fig. 1** Microglia do not phagocytose dendritic spines. Representative images of microglia (red, Cx3cr1::CreER; RC::LSL-tdTomato) **a** apposing or **b** encapsulating a dendritic spine (green, Thy1::EGFP; white insert: projection of the undeconvolved z-stack containing the contacted spine). Note that the neck of the contacted spine is intact (white arrow head). **c** Quantification of microglia–spine contacts. No spine was found phagocytosed, as contacted spines were always attached to their dendrite (n = 25 cells from 5 animals for 8944 spines analyzed, error bars are mean + SEM). **d** Encapsulated spine localizing next to a phagocytic compartment (blue, CD68 immunostaining). Scale bars: 0.5 µm

identified by confocal microscopy. Once a region of interest (ROI) was identified, concentric brands were etched into the surface of the fixed tissue with a UV-laser microdissector microscope and the tissue was embedded and prepared for electron microscopy. The resulting trimmed block of tissue was then subjected to X-ray imaging to identify vasculature and nuclear landmarks, and align these with the branding. The block was then subjected to FIB-SEM at 5 nm lateral pixel size and 8 nm section thickness using the landmarks as guides to capture the ROI. Four ultrastructural image stacks (up to $35 \times 20$ µm area; 2300 images) were obtained containing a total of 8 appositions and 5 encapsulations. Three-dimensional ultrastructural reconstruction confirmed microglia–spine apposition events where microglia and spine membranes were in contact or juxtaposed (7/8 apposed; Fig. 2f). However, most encapsulations appeared as simple appositions (3/5) with only one ROI showing > 50% of the spine surface contacted by microglia (Fig. 2b–f). These findings show that caution must be exercised when interpreting colocalization of synaptic material with microglia using light microscopy. Moreover, these data do not support the hypothesis that microglia phagocytose dendritic spine material.

**Microglia trogocytose presynaptic elements**. Previous studies have argued for the phagocytosis of presynaptic material by microglia[8,9,14,20] and we re-examined our electron microscopy datasets for evidence of microglia engulfment of identified presynaptic structures. We examined over 56 µm³ of microglial material for a total surface of 560 µm² from eight reconstructed cells. We found 17 confirmed double-membrane inclusion bodies (Fig. 3c). Two of these contained putative presynaptic vesicles as indicated by their 40 nm diameter (Fig. 3a, c and Supplementary Movie 2), suggesting that presynaptic

material is a substrate for elimination by microglia. In addition, we found 20 double-membrane structures that appeared to be in the process of engulfment. Many involved axonal shafts (8/20; Fig. 3b, c and Supplementary Movie 3) and a smaller number involved presynaptic boutons (3/20). We also frequently observed microglia self-engulfment in which a microglial process was enwrapped and pinched by another process from the same cell (9/20; Fig. 3c and Supplementary Fig. 2). Analysis of the size distribution of inclusions revealed that the material being engulfed from boutons and axons typically ranged between 0.01 and 0.05 µm³ (Fig. 3d) with an average diameter of $253 \pm 24$ nm. This shows that presynaptic structures are not entirely phagocytosed by microglia but rather "trogocytosed," a term originally coined to describe membrane transfer in immune cells[19] and later extended to refer to partial phagocytosis by various cell types, including macrophages[18,26,27]. We also observed numerous invaginations of microglia facing boutons or axons that allowed us to reconstruct the putative sequence of events leading to the microglial digestion of these structures (Fig. 3e). Engulfment did not appear to be mediated by the formation of phagocytic cups, as no microglial pseudopodia were observed at the contact site. Instead, boutons or axonal pinches appear to sink into microglial cytoplasm before closure of the membrane and subsequent trafficking. These findings argue for the specific trogocytosis of presynaptic structures by microglia and suggest that this activity may be oriented indiscriminately toward axons and synaptic boutons rather than selectively targeting the presynaptic active zone.

To explore the dynamics of interaction between synapses and microglia, we developed a time-lapse fluorescence imaging method in brain explant cultures (Supplementary Fig. 3a). Organotypic hippocampal slice cultures are known to undergo key developmental steps similar to those observed in vivo, including synapse maturation[28–31], and have been previously

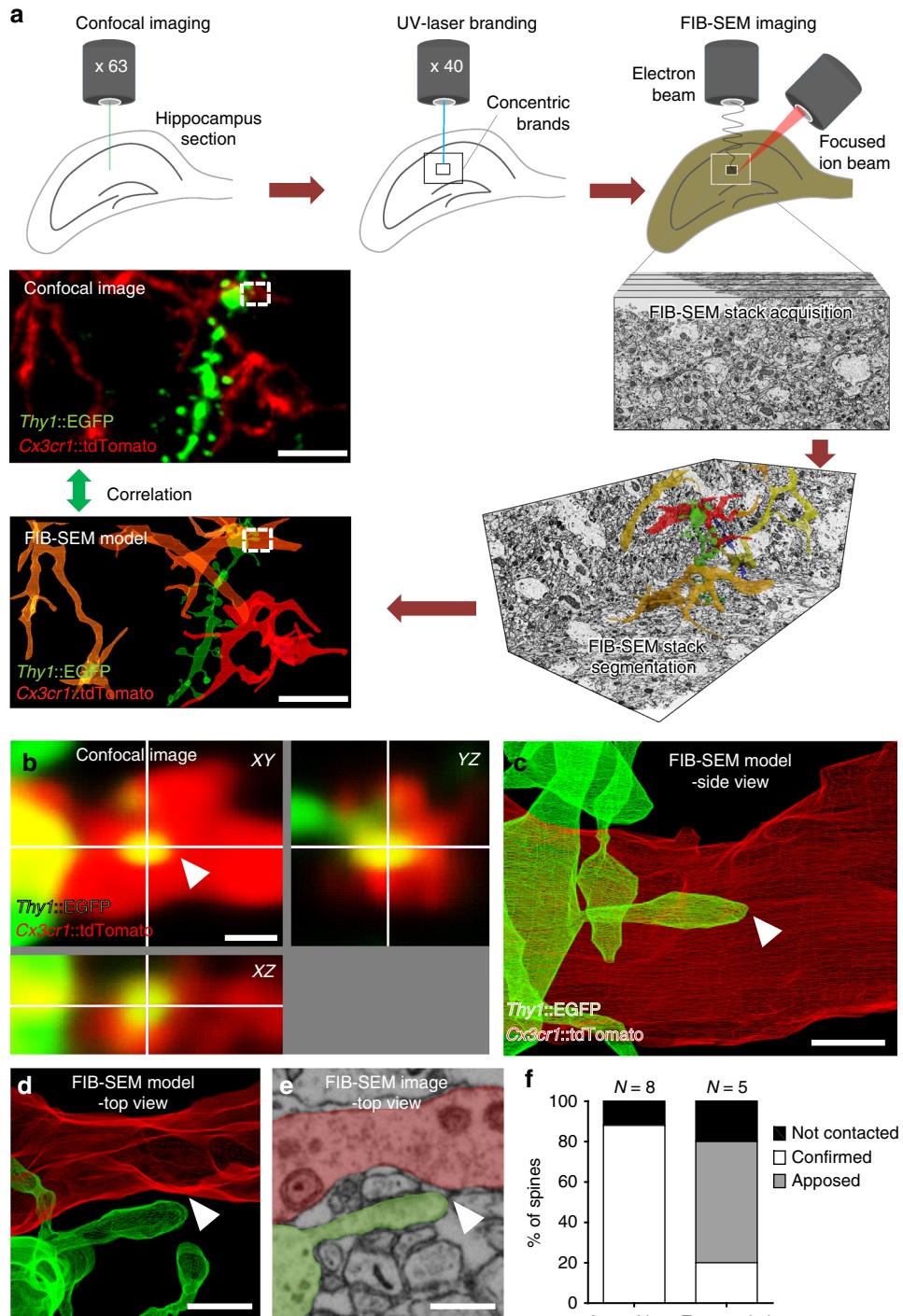

**Fig. 2** CLEM analysis of microglia-spine interactions. **a** Schematic of correlative light and electron microscopy (CLEM) workflow. **b** Confocal orthogonal view of a region of interest (ROI, dotted line in **a**) containing a spine encapsulated by microglia. **c** Segmentation of the ROI containing the encapsulation from the corresponding electron microscopy dataset, side view. **d** Top view of the ROI revealed that the spine was not encapsulated, with **e** no sign of elimination. **f** Quantification showed that the majority of the encapsulations observed by confocal microscopy were simple appositions by electron microscopy ($n = 13$ contacts analyzed from two animals). Scale bars: 5 µm for **a**, 0.5 µm for **b**–**e**

used to study ramified microglia function[32]. As shown previously, microglia initially respond to culturing by retracting their processes and assuming an activated phenotype[33] (Supplementary Fig. 3b). Following 1 week in culture, however, microglia morphology resembles that found in vivo[34] (Supplementary Fig. 3b, c). Time-lapse imaging of hippocampal cultures was performed using light sheet fluorescence microscopy, in order to minimize light toxicity common to point-source beam scanning

microscopes and to allow for the visualization of multiple fluorophores across very large fields of view (up to $0.5 \times 0.5 \times 0.2$ mm) at relatively high frame rates (up to 1 frame/45 s) for protracted periods (up to 3 h). To validate the technique and determine whether microglia in hippocampal explants showed in vivo-like physiology, we quantified the number and speed of process extension and retraction events (Supplementary Fig. 3d, e and Supplementary Movie 4). No significant difference was

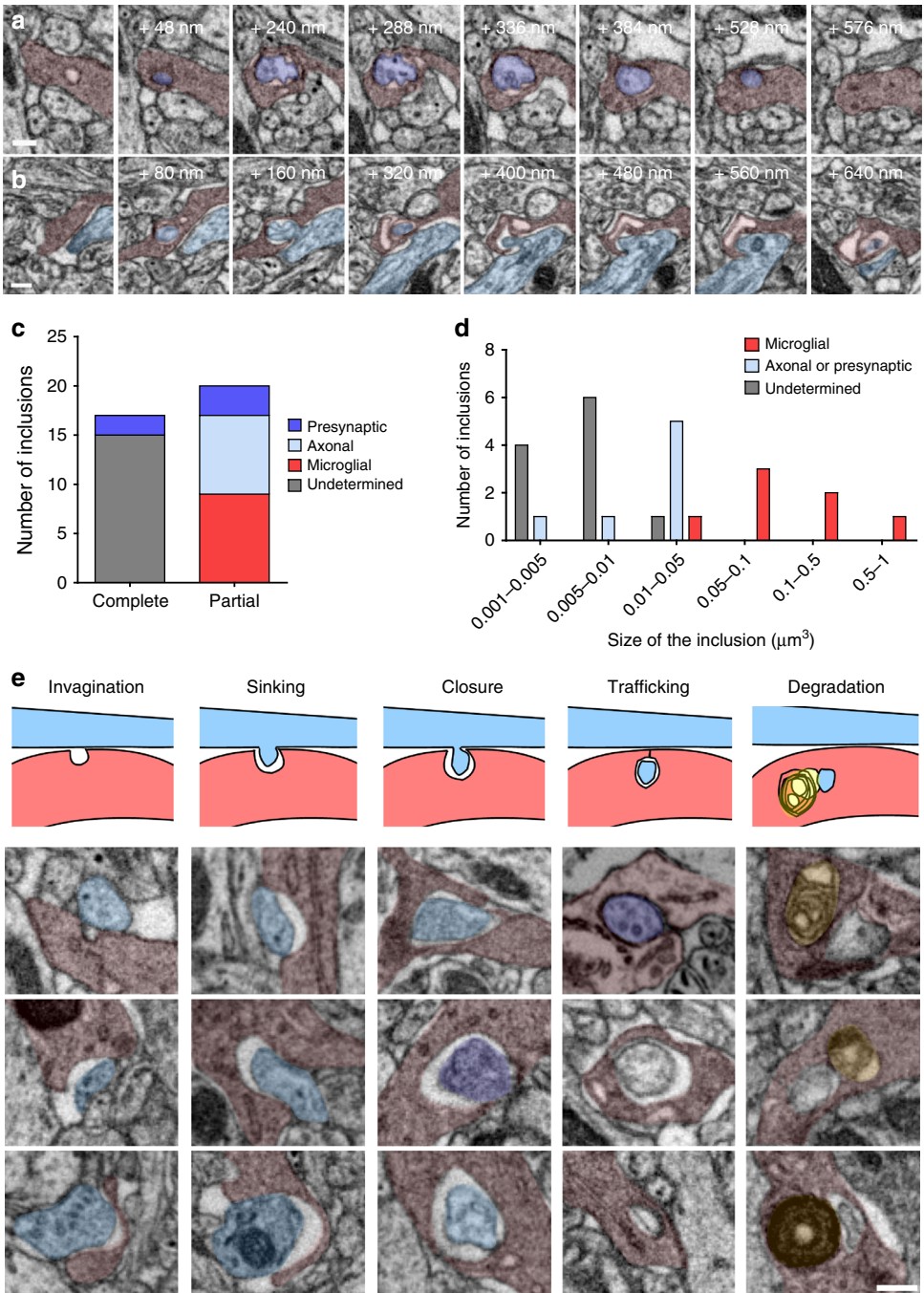

**Fig. 3** Microglia trogocytosis of presynaptic boutons and axons. Representative FIB-SEM image sequences of **a** complete presynaptic bouton inclusion (dark purple), as identified by its 40 nm vesicles content, inside microglia (red), **b** partial inclusion containing axonal material (clear blue) inside a microglia. **c** Quantification of microglial partial and complete inclusions ($n = 37$ inclusions from 8 cells, 4 animals). **d** Distribution of the volume of microglial inclusions. **e** Putative sequence of events leading to presynaptic bouton or axon material digestion by microglia, represented by a schematic and a collection of three examples for each step (gray: undetermined origin, yellow: lysosomes). Scale bars: 200 nm

observed between extension and retraction over time, with an average of 26 extension and 21 retraction events per cell per minute (Supplementary Fig. 3d, two-way analysis of variance (ANOVA), main effect of time: $F_{2, 12} = 0.22$, $p = 0.81$; main effect of direction: $F_{1, 12} = 0.96$, $p = 0.37$, $n = 4$ cells). The speed of extension and retraction events was similar and stable over time (1.9 and 1.8 µm/min, respectively; Supplementary Fig. 3e, two-way ANOVA, main effect of time: $F_{2, 12} = 3.48$, $p = 0.06$; main effect of direction: $F_{1, 12} = 0.10$, $p = 0.77$, $n = 4$ cells) and consistent with previous in vivo imaging studies[3]. Next, we

labeled presynaptic CA3 to CA1 Schaffer collateral projections with cytoplasmic near infra-red fluorescent protein (iRFP) following local adeno-associated viral infection (AAV-*Syn*::iRFP, Fig. 4a) of the CA3 region of organotypic slices from *Thy1*::EGFP; *Cx3cr1*::CreER; *RC*::LSL-tdTomato triple transgenic mice shortly after culturing. Importantly, microglia at the imaging site did not show any detectable morphological changes following viral infection, as they were imaged 2 weeks later and 500 µm distant from the infection site (Supplementary Fig. 4). Consistent with our fixed electron microscopy data, we found clear evidence for the

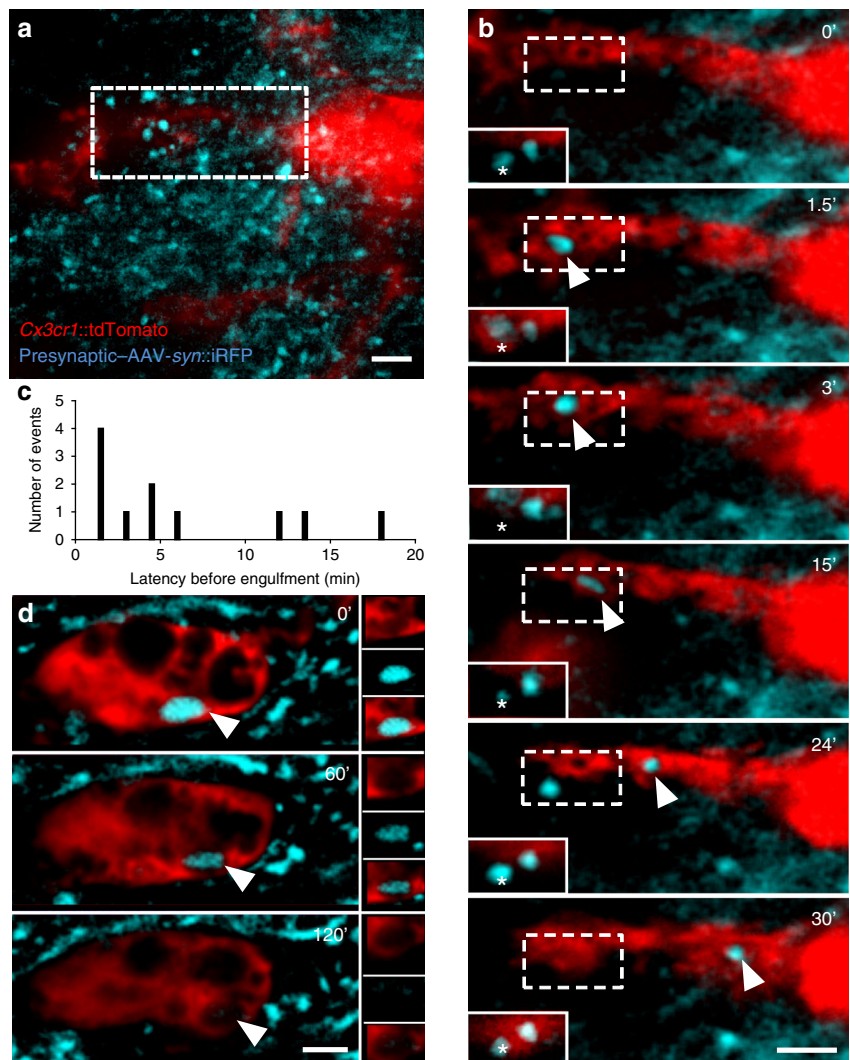

**Fig. 4** Rapid trogocytosis of presynaptic material by microglia. **a** Low-magnification image of a stack projection of 35 consecutive optical sections ($\Delta z =$ 0.48 μm) showing microglia (red, *Cx3cr1*::CreER; *RC*::LSL-tdTomato) surrounded by iRFP + presynaptic boutons from Schaffer collaterals (blue, AAV-*Syn*:: iRFP) in the CA1 region of organotypic hippocampal cultures. **b** Time-lapse imaging revealed engulfment of a presynaptic bouton (single optical planes series from **a**, dotted box). The corresponding optical plane containing the presynaptic bouton (star) is shown in the plain white insert. Although most of the bouton has been internalized by the microglia and trafficked toward the soma (arrowhead), presynaptic material remains at the original site (star), indicating partial elimination. **c** Distribution of the latency to engulfment ($n = 11$ events from 8 cells originating from 3 organotypic slice cultures). **d** Representative image of an iRFP + inclusion in a microglia soma (arrowhead) showing slow degradation. Scale bars: 2 μm

engulfment of presynaptic material (Fig. 4b, d and Supplementary Movie 5). Surprisingly, presynaptic engulfment events ($n = 11$ from 8 microglia analyzed) were rapid, frequently occurring in < 3 min (Fig. 4c), raising the possibility that some events occurring within our frame rate (1 frame/90 s) went unnoticed. Notably, even in those cases where microglia were seen to eliminate most of the synaptic bouton (2/11 events), presynaptic material remained at the initial site, confirming that microglia engage in partial elimination, or trogocytosis, of presynaptic boutons.

**Trogocytosis does not require CR3 signaling**. The complement system has been shown to be required for the engulfment of apoptotic cells by microglia in developing hippocampus[35], for the efficient pruning of synapses during retinothalamic development[6,9,36], and for the loss of synaptic structures during neurodegeneration and aging[37–40]. We therefore tested whether microglia trogocytosis of presynaptic elements was compromised in mice lacking the complement receptor CR3, an essential component of the

complement signaling pathway expressed on microglia. Using the time-lapse imaging setup previously described, we analyzed microglia–synapse interaction in slices from *CR3*-KO; *Thy1*::EGFP; *Cx3cr1*::CreER; *RC*::LSL-tdTomato quadruple transgenic mice (Fig. 5a). Contrary to our hypothesis, we found no evidence for a deficit in microglia trogocytosis in KO when compared with wild-type (WT) slices ($2.3 \pm 0.7$ vs $1.5 \pm 0.6$ trogocytosis events/cell for 3 h, respectively; $p = 0.37$, $t$-test; six and eight cells analyzed from three cultures, Fig. 5b, c and Supplementary Movie 6). There was also no difference in the latency of elimination (WT: $6.0 \pm 1.6$, KO: $3.9 \pm 1.1$, $p = 0.28$, $t$-test; Fig. 5d) or in the number of iRFP inclusions found in microglia soma (WT: $2.6 \pm 0.4$, KO: $2.2 \pm 0.6$ min, $p = 0.51$, $t$-test; Fig. 5e). These data suggest that the complement signaling pathway is not required for microglial trogocytosis of presynaptic elements.

**Microglia induce spine head filopodia formation**. To explore whether microglia might indirectly induce the elimination of spines as a consequence of non-phagocytic contact or presynaptic

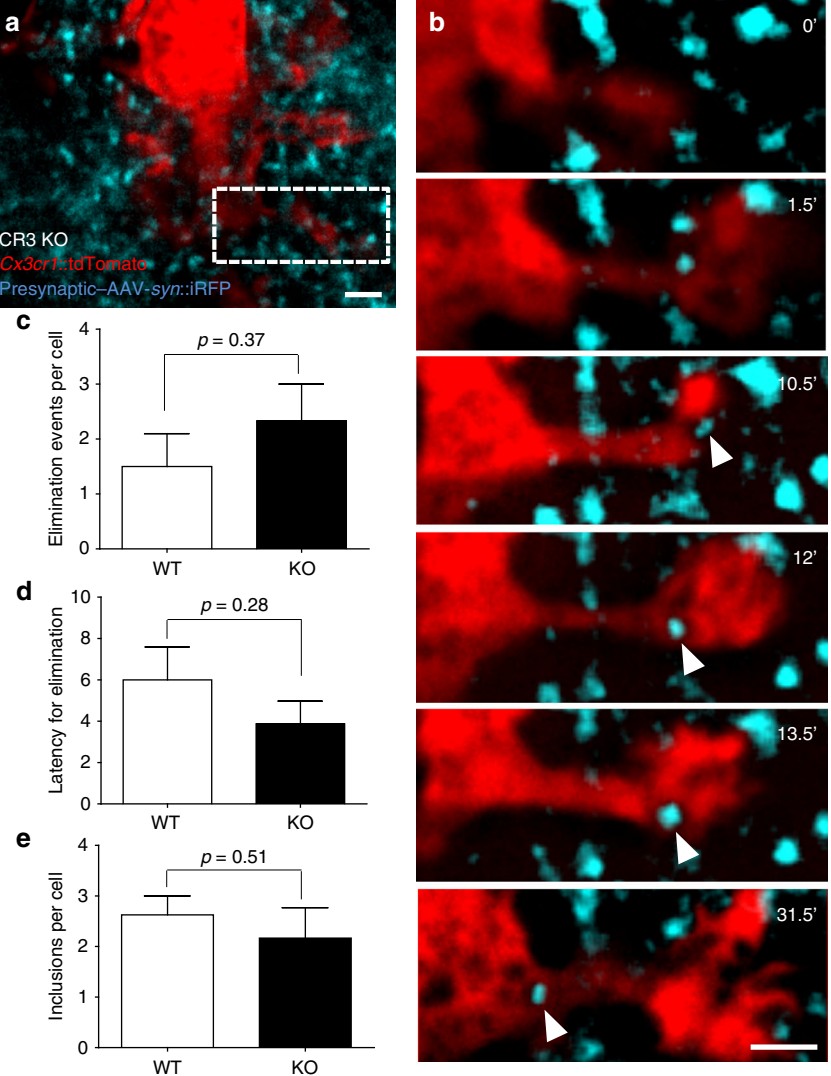

**Fig. 5** CR3 is not necessary for microglia trogocytosis. **a** Low-magnification image of a stack-projection of 33 consecutive optical sections (Δz = 0.48 μm) showing a microglia (red, *Cx3cr1*::CreER; *RC*::LSL-tdTomato) surrounded by presynaptic boutons from Schaffer collaterals (blue, AAV-*Syn*::iRFP) in organotypic hippocampal slices from CR3 KO mice. **b** Time-lapse imaging of CR3 KO microglia–bouton interactions revealed engulfment of presynaptic material (single optical planes series from **a**, dotted box). No difference was found in the **c** number or **d** latency of microglia engulfment events, or **e** the number of iRFP inclusions per cell between WT and CR3 KO slices (two-sided unpaired *t*-test, n = 8 and 6 cells from 3 organotypic slice cultures, error bars are mean + SEM). Scale bars: 2 μm

trogocytosis, we investigated microglia interactions with the postsynaptic compartment by time-lapse imaging in organotypic cultures. Putative contacts between microglia processes and spines were identified and analyzed over time (Fig. 6a). Microglia–spine contacts were brief (4.2 ± 0.85 min) and microglia frequently re-contacted the same spine, suggesting that the contacts were non-random. Ten percent of microglia-contacted spines (3/31) disappeared during the imaging session (Fig. 6b), and 13% both appeared and disappeared (4/31), and were classified as transient spines. However, none of these spines were in contact with microglia at the time of disappearance, arguing for a microglia-independent spine elimination process. Importantly, the frequency of disappearance of spines that had been contacted during the imaging session by microglia was not different from that of nearby (< 4 μm away), non-contacted spines (10% vs 11%, respectively, n = 28; Fig. 6c). Transient spines, on the other hand, were found exclusively among contacted spines when compared with nearby, non-contacted spines (13% vs. 0%). Closer, high-resolution inspection of these microglia-transient spine contact

events revealed that these spines formed from filopodia that appeared at the microglia contact point or in proximity to the dendritic shaft (4/31; Fig. 6a, d, i and Supplementary Movie 7) similar to a phenomenon recently observed in the mouse cortex using in vivo imaging[15]. Intriguingly, 39% (13/31) of the persistent, mature spines contacted by microglia formed a filopodium protruding from the head (spine head filopodia) and extending toward the microglia process (Fig. 6a, e, i and Supplementary movie 8), thus making spines a preferential substrate for microglia-induced filopodia compared to dendritic shaft (13 spine head filopodia vs. 4 shaft filopodia). Occasionally, we noted stretching of the entire spine during microglial contact, suggesting that the microglia process was able to induce profound changes in spine morphology (2/31; Fig. 6a, f, i). Systematic, morphometric analysis of microglia–spine contact events across the imaging session was used to perform a cross-correlation analysis that revealed a significant increase in spine length that peaked just after microglia contact (n = 13 spines analyzed, Fig. 6g). Vectorial correlation of spine head filopodia and

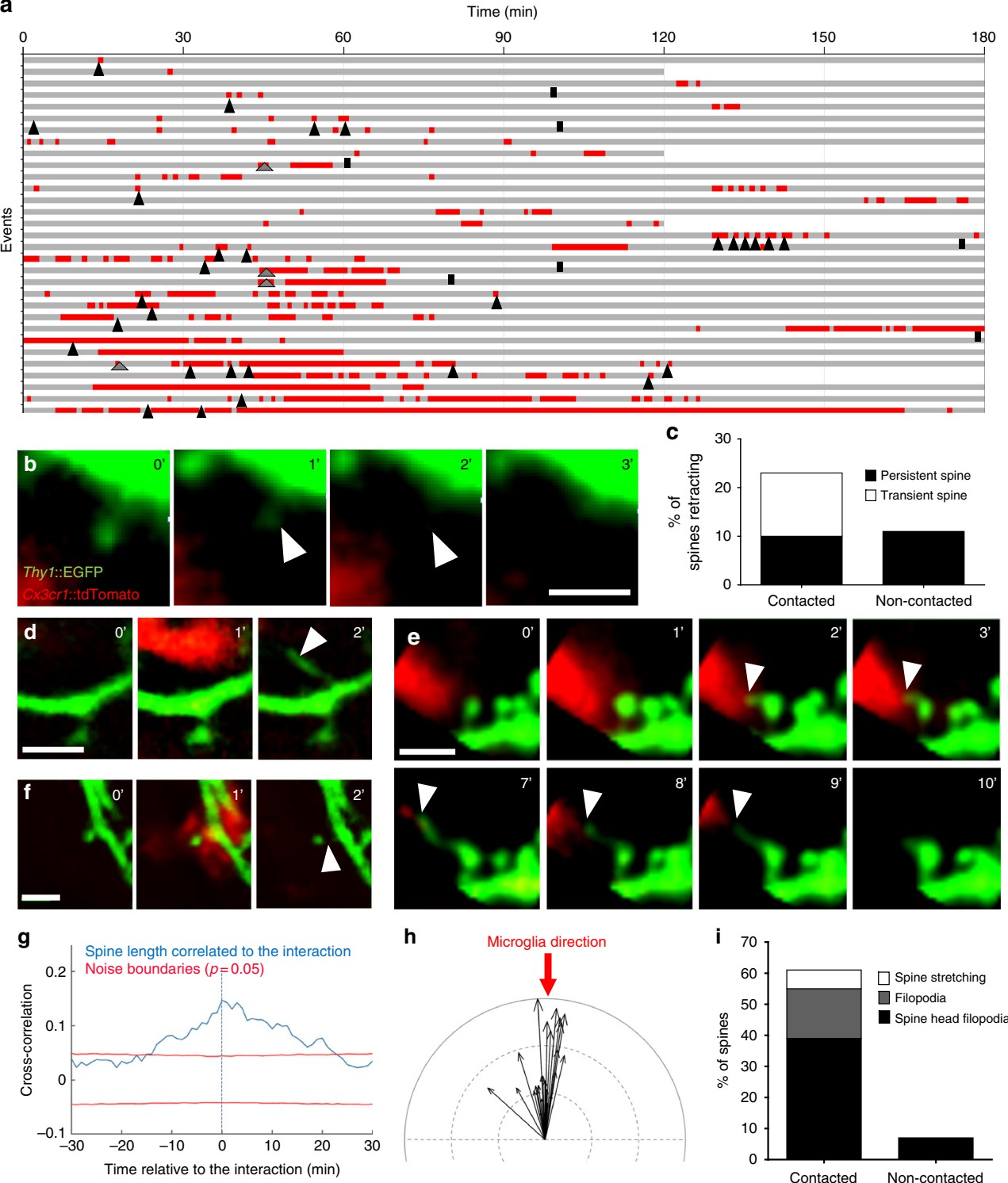

**Fig. 6** Microglia induce spine head filopodia formation. **a** Quantification of microglia-spine contact duration (red) over the imaging session (gray). Each line represents a spine selected to have been contacted at least once by microglia and annotated for spine appearance (gray arrowhead), disappearance (black bar), and spine head filopodia formation (SHF, black arrowhead). Representative time sequence images of **b** spine disappearance, **d** filopodia formation, **e** SHF, and **f** spine stretching. **c** Quantification of spine retraction rate of contacted versus non-contacted neighboring spine. **g** Cross-correlation analysis revealed a significant increase in spine length during microglia-spine contact. **h** Vectorial analysis showed a significant correlation of microglial process direction (red arrow) with filopodia direction (Anderson–Darling test, $P = 0.00047$, $n = 21$ spine head filopodia analyzed, black arrows indicating filopodia length and direction, indentation $= 1 \mu m$). **i** Quantification of spine stretching, filopodia formation, and spine head filopodia formation events in contacted versus non-contacted neighboring spines ($n = 31$ and 28, respectively, from 4 organotypic slice cultures). Scale bars: 2 μm

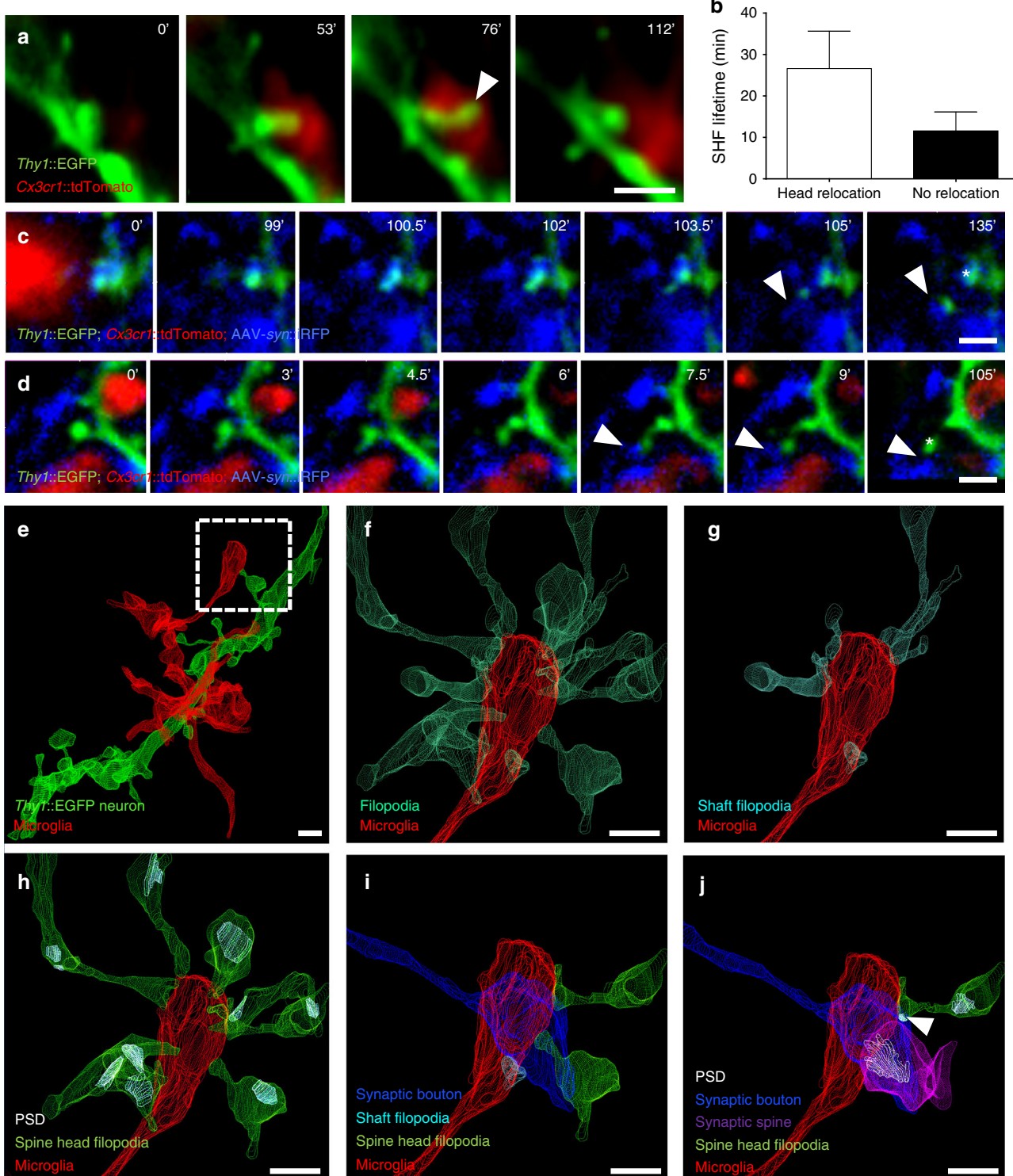

**Fig. 7** Spine head filopodia-associated synapse remodeling. **a** Representative time sequence images of a spine head relocating to the tip of the SHF following its induction by microglia. **b** Quantification of SHF lifetime revealed more stable filopodia following relocation ($n = 5$ relocating SHF vs 16 non-relocating, analyzed from 3 organotypic slice cultures, error bars are mean + SEM). **c,d** SHF making a stable contact with a neighboring bouton (arrowhead). In **c** the spine persists at its original location (star), whereas in **d** the spine relocates to the newly contacted bouton (star). **e** Example of microglial process in contact with a spine extending a SHF (dotted box) as identified and visualized by CLEM. Further examination of the fully segmented EM dataset revealed **f** multiple filopodia extending toward the microglial process, of which **g** a few were simple filopodia and **h** the majority originated from mature spines bearing postsynaptic-densities (PSDs). It is noteworthy that the microglial process is in intimate contact with a presynaptic bouton and **i** several of the SHFs contact the same bouton, **j** one of which has formed an immature PSD (arrowhead, Supplementary Fig. 5) resulting in the formation of a multiple-synapse bouton. Scale bars: 1 μm

microglia process movement direction confirmed that filopodia extended toward the microglia process, and varied in length from 0.4 to 3.1 µm with an average of 1.5 µm ($n = 21$ spine head filopodia formations analyzed, Fig. 6h). Notably, spine head filopodia were rarely found on nearby, non-contacted spines (7% on non-contacted spines vs 39% on contacted spines, $n = 28$ and 31 spines analyzed, respectively; Fig. 6i).

Spine head filopodia have been shown to contribute to the formation of new spine-bouton contacts[41] and proposed to be a mechanism for the movement, or "switching," of spines from one bouton to another, possibly in response to changing synaptic activity or plasticity[41,42]. Although our approach did not allow for a systematic assessment of such switching events, we did find that 27% (5/21) of microglia-induced spine head filopodia were associated with spine head relocation, as identified by the displacement of the head from its original site to the tip of the filopodia (Fig. 7a). Interestingly, spine head filopodia that underwent relocation showed a tendency for longer lifetimes than those that did not (27 vs 12 min, $n = 5$ and 16, respectively; Fig. 7b), suggesting that this relocation might be associated with stabilizing synapse formation. This hypothesis was supported by two cases in which we were able to simultaneously image GFP + spines and iRFP + presynaptic boutons, and we could confirm that the induced spine head filopodia made stable contact with a different, neighboring bouton (Fig. 7c, d).

A systematic analysis of our FIB-SEM datasets allowed us to confirm the frequent presence of microglia-associated spine head filopodia in fixed brain tissue and rule out that this phenomenon was an artifact of the organotypic culture. Remarkably, over 28% (5/18) of mature, PSD-containing spines found in contact with microglia processes presented a filopodium extending toward microglia. In one particularly striking case (Fig. 7e), a microglial end process contacting a presynaptic bouton was found surrounded by converging filopodia (15 filopodia originating from 9 dendrites; Fig. 7f). Consistent with our live-imaging data, the majority (9/15) were spine head filopodia originating from mature, PSD-containing spines (Fig. 7h), whereas the rest were filopodia extending from dendritic shaft structures (6/15; Fig. 7g). Intriguingly, a few of these spine head filopodia extended alongside the microglia-contacted presynaptic bouton (Fig. 7i) and one appeared to initiate a synapse as indicated by the presence of a PSD but no clustering of presynaptic vesicles, resulting in the formation of a multiple-synapse bouton (MSB, Fig. 7j and Supplementary Fig. 5). We also noted that 12/15 of the converging filopodia shared their dendrite of origin, suggesting that they could facilitate the formation of class I MSBs in which one bouton synapses with several spines originating from a common neuron, a form of synaptic contact that conveys increased efficacy[43]. MSB1s have previously been shown to depend on microglia–neuron signaling and contribute to the strengthening of developing hippocampal circuits and the establishment of normal brain connectivity[11]. Together, these data suggest that microglia are broadly involved in structural synaptic plasticity and circuit maturation, both by the trogocytosis of presynaptic boutons and axons, as well as by the induction of filopodia from postsynaptic sites (Supplementary Fig. 6).

## Discussion
Our findings confirm the hypothesis that microglia directly engulf and eliminate synaptic material. However, contrary to previous assumptions, we found no evidence for the phagocytosis of entire synapses. Instead, we observed microglia trogocytosis—or nibbling—of synaptic structures. Importantly, microglia trogocytosis was restricted to presynaptic boutons and axons, with no evidence for elimination of postsynaptic material. Intriguingly, microglia contacts at postsynaptic sites frequently elicited transient filopodia,

most of which originated from mature spines. These data support the current hypothesis that microglia can "eat" synaptic material, but point to a more nuanced role for microglia in synapse remodeling that may explain the diverse synaptic alterations observed following the disruption of microglial function.

Our observation of microglia engulfment of presynaptic material is consistent with published reports showing the localization of material deriving from axonal projections inside microglia[9]. To the best of our knowledge, our data are the first time-lapse images to directly demonstrate the active engulfment of synaptic material by microglia. Our extensive characterization of microglial content from three-dimensional (3D) FIB-SEM reconstructions validate previous data from single section electron microscopy that showed putative double-membrane inclusions of presynaptic material in microglia[9]. Moreover, our observation of intermediates such as invaginations, pinching of presynaptic boutons and axons, and complete inclusions sheds light on the cellular mechanism involved. Synapses were previously proposed to be eliminated by microglia through phagocytosis, a process traditionally defined as the cellular uptake of particles over 0.5 µm in size[44]. Instead, our data show that only small fragments (250 nm average diameter, Fig. 3) of the presynaptic compartment are engulfed by microglia. This partial elimination, or trogocytosis (from the Greek trogo: to nibble) has been previously described in immune and amoeboid cells[18,26,27] that ingest small parts (< 1 µm) of their targets within a few minutes, a timeframe compatible with our observations (Fig. 4). Although phagocytosis and trogocytosis likely share common endocytic machinery, they potentially differ in their uptake pathways. In fact, we found that CR3, a microglia-expressed complement receptor involved in phagocytosis and previously proposed to mediate synapse elimination in the retinogeniculate pathway[9], is not necessary for the trogocytosis of presynaptic structures. However, other components of the complement pathway could be involved and it is possible that different brain regions recruit different pruning pathways. One possible candidate "eat me" signal is phosphatidylserine (PS), a phospholipid known to elicit engulfment by macrophages following its exposure on the outer leaflet of the cell membrane[45] and recently suggested to mediate trogocytosis[27]. The known capacity of PS to laterally diffuse within the membrane could explain our observation that microglia indiscriminately trogocytose presynaptic boutons and axonal shafts. It should also be noted that the elimination of presynaptic material does not necessarily imply elimination of functional synaptic structures, and our observation that microglia trogocytosed primarily axonal shafts rather than boutons suggests that this process may be relatively nonspecific. Presynaptic trogocytosis may be aimed at an overall reduction in axonal processes or, alternatively, it may mediate a remodeling of axons by eliminating, e.g., specific surface-associated factors that inhibit presynaptic site formation.

Second, several data argue against a major role for microglia in the elimination of postsynaptic material in the developing hippocampus, including the following: (1) absence of phagocytosis/trogocytosis of postsynaptic material by microglia, as assessed by confocal microscopy in fixed brain sections (Figs. 1–2), (2) the presence of frequent and intimate partial inclusion events ("pinched" material) of exclusively presynaptic boutons and axons as assessed by 3D volume EM (Fig. 3), and (3) intake of presynaptic (Fig. 4) but not postsynaptic (Fig. 6) material by microglia as assessed by time-lapse fluorescence imaging. The fact that microglia do not eliminate postsynaptic structures in the developing hippocampus has implications for the interpretation of previous data in the field. For example, we and others have published data showing localization of postsynaptic proteins inside microglia[8,14,20,46]. In our current confocal microscopy data, we observed the intimate encapsulation of spines by

microglia, a phenomenon that likely explains the previously described colocalization of immunolabeled postsynaptic proteins with microglia. However, using cytoplasmic labeling we noted that all contacted spines were still attached to the dendritic shaft via a spine neck and our 3D FIB-SEM reconstructions did not support the phagocytic engulfment of postsynaptic material by microglia. Previous evidence for the localization of postsynaptic material inside microglia by single-section electron microscopy, either by contrast agent-enhanced visualization of PSD-like material[20] or its immunodetection[8], are potentially more difficult to counter, but might reflect the limitations inherent to two-dimensional electron microscopy. In addition, it should be noted that PSD95 has been found to be expressed by microglia[47] (but see ref. [48]), and that immunocolocalization of synaptic proteins with microglia might derive from non-phagocytic membrane exchange[49]. Although it remains possible that elimination of postsynaptic material by microglia might occur in other brain regions, at other developmental stages, or alternatively at such a low frequency that it went unnoticed in our experiments, our data argue that immunofluorescent colocalization of postsynaptic proteins and microglia, even by super-resolution methods (e.g., stimulated-emission depletion), should be interpreted with caution.

Lastly, a major observation from our time-lapse imaging data was the induction of transient filopodia following microglia contacts. This observation is in line with a recent in vivo imaging study reporting that microglia contacts can induce local calcium transients in dendritic shafts followed by filopodia formation[15]. It is also consistent with a report showing that microglia participate in the learning-dependent formation of functional synapses via brain-derived neurotrophic factor (BDNF) and its receptor TrkB[16]. Together, these observations argue that filopodia induction by microglia may be a widespread phenomenon and a possible trigger mechanism for the formation of functional synapses. Intriguingly, we observed that the majority of filopodia induced by microglia originate from mature spine heads (Fig. 6), a finding confirmed in our electron microscopy datasets. Protrusions from spine heads have been previously described in the hippocampus[41,50–55], in the visual cortex[56], and in the olfactory bulb[42], where they have been referred to as spinules, spine head protrusions, or spine head filopodia, depending on the length of the extension measured. Spine-emerging spinules[54] were considered trans-endocytic and surrounded either by axonal material or astrocytes, the later of which might relate to astrocyte-mediated synaptic pruning described in the thalamic system[57]. On the other hand, spine head filopodia or protrusions were found in the extracellular space between cells, and associated with microtubule invasion of the spine[58] and synaptopodin-dependent actin bundling[51]. The mechanism for spine head filopodia induction by microglia in our preparations is not clear, but might involve the application of tension to "pull" the filopodia, a loosening up of the extracellular matrix by microglia invasion, or a release of chemotactic factors. Intriguingly, BDNF has been shown to induce spine head filopodia[42], as well as synapse formation upon expression by microglia[16], suggesting a role for this factor in microglia-dependent circuit remodeling during development.

Our time-lapse imaging study revealed that microglia-induced spine head filopodia formation was frequently followed by a relocation of the spine head to the tip of the filopodium and occasionally resulted in a new bouton contact (Fig. 7). These observations confirm that spine head filopodia induction might trigger spine switching and potentially the replacement of inefficient with more efficient synapses[41,42,53]. Spine head filopodia can be induced by neurotransmitter such as acetylcholine[53] and glutamate via the activation of AMPA receptors[41,42,55]. Moderate, but not high, levels of glutamate induce spine head filopodia formation[41,52], opening the possibility that microglia-mediated spine

head switching may be moderated by glutamatergic neuro-transmission and presynaptic release probability, a hypothesis supported by observation that blockade of evoked neurotransmitter release by the application of tetrodotoxin increased the incidence of spine head filopodia[41]. Spine switching has the potential to transform single-synapse boutons into MSBs[41,59] and if these occur on spines emerging from the same dendrite they might generate so called class I MSBs, a subclass of excitatory connections that increase during postnatal development and mediate the strengthening of excitatory connectivity[11,43,50]. Further evidence to support a role for microglia-induced spine head filopodia in MSB formation comes from our anecdotal observation of a dozen spine head filopodia converging toward a single microglia process in intimate contact with a presynaptic bouton on which they formed a nascent MSB (Fig. 7 and Supplementary Fig. 5). Although caution must be exercised when extrapolating such anecdotal electron microscopy data, the architecture of this particular case where most of the filopodia shared their dendrite of origin supports the hypothesis that microglia might promote the formation of class I MSBs. Such a hypothesis is consistent with previous observations that mice lacking the microglia–neuron signaling factor fractalkine (Cx3cl1) are associated with a deficiency in class I MSBs and impaired maturation of functional circuit connectivity[11]. Overall, our data argue that the prevailing view of microglia as phagocytic cells eliminating synapses during neural circuit development may be overly simplified. Instead, they suggest a broad role for microglia in synaptic remodeling via the trogocytosis of axonal structures, and the induction and reorganization of postsynaptic sites, so as to achieve an appropriate maturation of circuits.

## Materials and methods

**Animals**. C57BL/6J mice were obtained from local EMBL colonies. *Thy1*::EGFP; *Cx3cr1*::CreER; *RC*::LSL-tdTomato triple transgenic mice were obtained by crossing *Thy1*::EGFP-M[21] (Jackson Laboratory stock 007788) with *Cx3cr1*::creER-YFP[16] (Jackson Laboratory stock 021160) and *Rosa26-CAG*::loxP-STOP-loxP-tdTomato-WPRE[22] (Jackson Laboratory stock 007905). Mice were used in homozygous state for *Thy1*::EGFP and in heterozygous state for *Cx3cr1*::CreER and *RC*::LSL-tdTomato. Cre-mediated recombination was induced by a single injection of 98% Z-isomers hydroxy-tamoxifen diluted in corn oil at 10 mg/mL (Sigma, 1 mg injected per 20 g of mouse weight) at P10. Residual yellow fluorescent protein (YFP) expression in microglia yielded a faint signal in GFP channel that was thresholded out in all analysis. For CR3-KO experiments, triple transgenic mice were additionally crossed with CD11b-deficient mice[60] (Jackson Laboratory stock 003991) and used in homozygous state. *Cx3cr1*::GFP mice[61] (Jackson Laboratory stock 005582) were used in heterozygous state. All mice were on a C57BL/6J congenic background. Mice were bred, genotyped, and tested at EMBL following protocols approved by the Italian Ministry of Health.

**Analysis of microglia phagocytic capacity**. C57BL/6J WT mice were anesthetized intraperitoneally with 2.5% Avertin (Sigma-Aldrich, St Louis) and perfused transcardially with 4% paraformaldehyde (PFA) at P8, P15, P28, and P40. Brains were removed and post-fixed in 4% PFA overnight (ON) at 4 °C. Coronal 50 μm sections were cut on a vibratom (Leica Microsystems, Wetzlar, Germany) and blocked in 20% normal goat serum and 0.4% Triton X-100 in phosphate-buffered saline (PBS) for 2 h at room temperature. CD68 and Iba1 were immunodetected by ON incubation at 4 °C with primary antibodies (rat anti-CD68 1:500, Serotec; rabbit anti-Iba1 1:200, Wako) followed by secondary antibodies (goat anti-rabbit A647 and goat anti-rat A546, 1:400, Life Technologies) incubation in PBS with 0.3% Triton-X100 and 5% goat serum for 2 h at room temperature. Sections were imaged on a TCS SP5 resonant scanner confocal microscope (TCS Leica Microsystems, Mannheim) with a × 63/1.4 oil-immersion objective at 48 nm lateral pixel size with an axial step of 130 nm. Iba1-positive microglia were 3D reconstructed using local contrast on Imaris software and CD68 signal intensity was measured in each individual reconstructed cell. CD68 expression pattern observed in Iba1-labeled microglia was confirmed in genetically labeled microglia (*Cx3cr1*::CreER; *RC*::LSL-tdTomato).

**Characterization of microglia–spine interactions**. Brain tissue was collected at P15 as previously described. Sections were permeabilized with PBS and 0.5% Triton X-100 for 30 min, and blocked with PBS, 0.3% Triton, and 5% goat serum for 30 min at room temperature. CD68 was immunodetected by ON incubation at 4 °C with primary antibodies (rat anti-CD68 1:500, Serotec) followed by secondary antibodies (goat anti-rat A647, 1:600, Life Technologies) incubation in PBS with 0.3% Triton and 10% goat serum at 4 °C ON. Secondary dendrites of bright GFP + neurons were

imaged in medial stratum radiatum of CA1 using Leica SP5 confocal resonant scanner microscope with a ×63/1.4 oil-immersion objective, at a lateral pixel size of 40 nm and an axial step of 130 nm. Images were deconvolved using Huygens software (40 iterations, 0.1 of quality change, theoretical point spread function) and sharpened using Image J software (NIH). Interactions were determined after 3D visualization in Imaris as follows: appositions were considered when 20–50% of the spine head surface was covered by microglia, encapsulation when > 70% was covered.

**Imaging microglia–spine interactions for CLEM**. Mice were perfused transcardially with PBS and fixed with 2% (w/v) PFA, 2.5% (w/v) Glutaraldehyde (TAAB) in 0.1 M phosphate buffer (PB) at P15. After perfusion brains were dissected and postfixed in 4% PFA in PB 0.1 M ON at 4 °C. Subsequently, 60 µm-thick vibratome (Leica Microsystems) coronal sections were cut and 4′,6-diamidino-2-phenylindole (DAPI) stained. Hippocampal areas were trimmed and mounted with 1% Low Melting Agarose (Sigma) in PB 0.1 M on glass-bottom dishes with alpha numeric grid (Ibidi). ROIs containing microglia–spine interactions were imaged at high magnification with TCS SP5 resonant scanner confocal microscope with a ×63/1.2 water-immersion objective, at a pixel size of 48 nm and a step size of 300 nm. Low-magnification stacks containing the ROI were acquired in bright field, GFP, RFP, and DAPI channels to visualize neurons and microglia together with fiducial capillaries and cell nuclei. A UV-diode laser operating at 405 nm, an Argon laser at 488 nm, and a diode-pumped solid-state laser at 561 nm were used as excitation sources. Subsequent to confocal imaging, the grid-glass bottom was separated from the plastic dish and placed onto Laser capture system microdissector microscope (Leica LMD7000) for laser etching of the ROI. The etched sections were retrieved and stored in 4% PFA in PB 0.1 M at 4 °C. Apposition/encapsulation events in Fig. 2 derive from four ROIs containing four neurons from a total of two animals. Microglial content analysis in Fig. 3, on the other hand, derives from seven ROIs containing end processes from eight microglia from a total of four animals.

**Sample preparation for FIB-SEM**. The sections were processed as described in Maco et al.[24]. Briefly, the sections previously imaged on confocal were washed in cold sodium cacodylate buffer 0.1 M pH 7.4, postfixed with 1% OsO₄/1.5% potassium ferrocyanide for 1 h on ice, followed by a second step of 1 h in 1% OsO₄ in sodium cacodylate buffer 0.1 M pH 7.4 on ice. Samples were then rinsed carefully in water and stained "en block" with 1% aqueous solution of uranyl acetate ON at 4 °C, dehydrated with raising concentration of ethanol and infiltrated in propylene oxide/Durcupan mixture with increasing concentration of resin. Durcupan embedding was carried out in a flat orientation within a sandwich of ACLAR® 33 C Films (Electron Microscopy Science) for 72 h at 60 °C. Sections were washed in cold sodium cacodylate buffer 0.1 M pH 7.4, postfixed with 2% OsO₄/1.5% potassium ferrocyanide for 1 h on ice, followed by a step with thiocarbohydrazide for 20 min at room temperature and then a second step of 30 min in 2% aqueous OsO₄ on ice. Samples were then rinsed carefully in water and stained "en block" first with 1% aqueous solution of uranyl acetate ON at 4 °C and then with lead aspartate at 60 °C for 30 min. Subsequently, sections were dehydrated with increasing concentration acetone and infiltrated in Durcupan resin ON followed by 2 h embedding step with fresh resin. As a pilote experiment, one of the samples (Zeiss, Oberkochen, Fig. 2) was processed using a slightly modified protocol based on the application of heavy metal fixatives, stains, and mordanting agents[62] producing slightly more contrast compared with the other samples.

Flat embedded samples were then trimmed to about 1 mm width to fit on the pin for microscopic X-ray computed tomography (MicroCT). Samples were attached to the pin with either double sided tape or dental wax and mounted into the Brukker Skyscan 1272 for microCT imaging. Data were acquired over 180° at a pixel resolution of 1.5–2 µm. Karreman et al.[25] thoroughly details the process of how the MicroCT data enable the correlation of fluorescent imaging to 3D electron microscopy of voluminous samples. In this experiment, MicroCT revealed laser etched markings performed at the microdissector microscope, and vasculature. This vasculature, which could also be seen by negative contrast in confocal datasets, acted as fiducial features to register the various microscopy modalities (MicroCT, and low- and high-magnification confocal data). Using Amira software (FEI Company), 3D models were generated from these microscopy modalities by thresholding and manual segmentation. These volumes could then be registered together by a manual fit to reveal the position of the event visualized by fluorescent confocal microscopy despite the loss of fluorescence during processing for EM. The registered volumes also allowed precise trimming of the sample for FIB-SEM, where it is necessary for the ROI to be at the surface of the sample or within 5 µm (for trimming procedure see Karreman et al.[25]). Each sample was trimmed according to the available features that would assist with later steps of FIB-SEM acquisition, such as to position the platinum deposition on the trimmed sample surface over the ROI and to position the imaging area on the cross-section face. For example, the laser markings that were made on one surface of the brain slice gave us only the ROI position in x but not over the thickness of 60 µm brain slice. For this axis, patterns made by the distribution of the vasculature were necessary to pin-point the position of the protective platinum coat over the ROI.

**FIB-SEM imaging**. Registered and trimmed samples were then mounted onto the edge of an SEM stub (Agar Scientific) with silver conductive epoxy (CircuitWorks) with the trimmed surface facing up so that it will be perpendicular to the focused ion

beam (FIB). The sample was then sputter coated with gold (180 s at 30 mA) in a Quorum Q150RS coater before being placed in the Zeiss Crossbeam 540 focused ion beam scanning electron microscope (FIB-SEM). Once the ROI was located in the sample, Atlas3D software (Fibics Inc. and Zeiss) was used to perform sample preparation and 3D acquisitions. First, a platinum protective coat of 20 × 20 µm was deposited with 1.5 nA FIB current. The rough trench was then milled to expose the imaging cross-section with 15 nA FIB current, followed with a polish at 7 nA. Now that the imaging cross-section was exposed, the features visible here including vessels and nuclei were used to correlate with the registered 3D volumes in Amira and confirm the current position relative to the ROI. During the acquisition, lower resolution keyframes with a large field of view (FOV) from 40 × 40 to 70 × 70 µm were acquired, in order to have this broader context of the sample. Provided there were enough features close to the ROI, this information helped to position the high-resolution imaging FOV (typically 10 × 10 µm). The 3D acquisition milling was done with 3 nA FIB current. For SEM imaging, the beam was operated at 1.5 kV/700 pA in analytic mode using the EsB detector (1.1 kV collector voltage) at a dwell time from 6 to 8 µs with no line averaging over a pixel size of 5 × 5 nm and slice thickness 8 nm. For the pilot acquisition run at Zeiss, Oberkochen, a large volume was first acquired at low magnification without prior MicroCT, while correlating with the confocal dataset to detect the ROI, which was subsequently imaged at 5 nm isotropic pixel size.

**FIB-SEM stack segmentation and correlation**. A single stack file containing individual FIB images was aligned on ImageJ software (https://imagej.nih.gov/ij/) with the help of the linear stack registration plugin (SIFT). Grayscale look up table was inverted and the stack was binned 2 × in both lateral and axial planes. Microglia and dendrites of interest were located based on their xyz coordinates within the ROI and from fiducials correlated between electron and confocal microscopy datasets. The segmentation was carried out manually using iMOD software (http://bio3d.colorado.edu/imod/) and 3D model was generated and matched to the confocal dataset to confirm perfect correlation. Thy1::EGFP neuron and microglia were identified upon correlation of light and electron microscopy datasets. Complete presynaptic bouton inclusions were identified by the presence of 40 nm vesicles typical of presynaptic machinery. Partial presynaptic bouton and axonal material inclusions were identified after 3D reconstruction of the belonging axon, based on the presence of presynaptic machinery: clusters of 40 nm presynaptic vesicles and apposition on a PSD. Blender software (https://www.blender.org/) was used to generate animation of the FIB-SEM dataset segmentation and 3D reconstruction.

**Preparation of hippocampal slice culture**. Organotypic hippocampal slice cultures were prepared using the air/medium interface method[63]. Briefly, mice were decapitated at P4 and hippocampi were dissected out in cold dissecting medium (Hank's buffered salt solution 1 × (Gibco), penicillin/streptomycin 1 ×, HEPES (Gibco) 15 mM, glucose (Sigma) 0.5%)). Transverse sections of 300 µm thickness were cut using a tissue chopper. Slices were laid on culture-inserts (Millipore) in pre-warmed six-well plates containing 1.2 mL of maintaining medium (minimal essential medium (MEM) 0.5 × (Gibco), Basal Medium Eagle 25%, horse serum 25%, penicillin/streptomycin 1 ×, GlutaMAX 2 mM, glucose 0.65%, sodium bicarbonate 7.5%, ddH₂O qsp). Medium was replaced after 24 h and then every 2 days. For culture of Thy1::EGFP; Cx3cr1::CreER; RC::LSL-tdTomato brain slices, 98% Z isomers-OHT was added to the maintaining medium at 0.1 µM during the first 24 h. After preparation, cultures were maintained for up to 21 days in vitro (DIV21) in incubator at 35 °C and 5% CO₂. The morphology of microglia was inspected in Cx3cr1::GFP slices at the indicated time-points upon fixation in PFA 4% for 1 h at room temperature, PBS-washed, mounted with mowiol, and imaged on a Leica SP5 confocal resonant scanner microscope with a 63 × /1.4 oil-immersion objective.

**Light sheet live imaging**. Live imaging of microglia and synapses in hippocampal slice cultures (DIV10–19) was performed using a Z1 light sheet microscope (Zeiss). The imaging chamber was set at 35 °C and 5% CO₂, and filled with imaging medium (MEM without phenol red 0.5 ×, horse serum 25%, penicillin/streptomycin 1 ×, GlutaMAX 2 mM, glucose 0.65%, sodium bicarbonate 7.5%, ddH₂O qsp) 2 h before the imaging session, to allow the system to equilibrate and the medium to reach pH7. Low melting point agarose was prepared at 2% with imaging medium and incubated at 35 °C and 5% CO₂ for 30 min, for reaching proper pH. The membrane of the Millipore insert containing the slice of interest was cut around the slice and laid onto the equilibrated liquid agarose before polymerization at 4 °C for 1 min. Although it is very unlikely that the tissue temperature dropped to this temperature, it has been reported that following incubation at 4 °C for longer periods of time (> 30 min) microglia soma enlarged, but recovered normal morphology within 2 h[64]. Therefore, we waited 2 h for the slice to recover before imaging and carried out control experiments showing that microglia motility (Supplementary Fig. 3) was similar to that found by two-photon imaging in vivo. The slice mounted on agarose was then placed in incubation at 35 °C and 5% CO₂ for further 30 min. The polymerized agarose containing the slice was then inserted into a FEP tube adapted on a glass capillary and the slice was gently pushed to be exposed 1 mm outside the FEP tube. The capillary was then placed on the microscope holder and the slice was immersed in the imaging medium of the chamber 1 h before imaging for stabilization. For all imaging sessions, microglia

and neurons were selected for their brightness and position in the stratum radiatum of CA1, 2 to 30 μm from the slice surface. Imaging was performed for 2–3 h using a 60× /NA1 water-immersion objective, with a lateral pixel size of 130 nm and an axial step of 480 nm. For microglia–postsynaptic structures interactions (2-colors imaging), 488 and 561 lasers were used for simultaneous acquisition of GFP and tdTomato signal using 505–545 and 575–615 band-pass filters on two cameras, at a rate of one frame/45–60 s. For microglia-pre/postsynaptic structures interactions (3-colors imaging), two channels were fast switching between frames: 488 and 561 lasers were used for simultaneous acquisition of GFP and tdTomato signal using a 505–545 band-pass filter and a 585 long-pass filter, and iRFP was imaged using a 638 laser line and the 585 long-pass filter, at a rate of 1 frame/90 s. The emission spectra of tdTomato and iRFP overlap significantly and are both efficiently detected in the presence of the 585 nm long-pass filter. However, we were able to image them separately, because their excitation spectra are distinct, with the 561 nm laser exciting primarily tdTomato (< 20% iRFP peak excitation) and the 638 nm laser exciting exclusively iRFP. This allowed us to image iRFP and tdTomato separately by alternating exposure to 561 and 638 nm light sources with a fixed 585 nm long-pass filter detection system. Notably, no detectable iRFP signal was seen during 561 nm illumination, most likely to be because iRFP is significantly more dim than tdTomato. All datasets were deconvolved using Zen software, and corrected for drifts on Image J using a script created by Albert Cardona and Robert Bryson-Richardson[65], and modified by Cristian Tischer (EMBL Heidelberg).

**Analysis of microglia motility**. TdTomato signal intensity was measured in microglial processes and normalized across all datasets. Noise was measured outside microglia, and removed by thresholding to the measured value + 40%. Motility was assessed by analyzing protrusions over 1 min interval. Extending and retracting protrusions were counted and measured at 0, 60, and 120 min after the beginning of the session to confirm imaging.

**Labeling of presynaptic structures**. AAV-rSyn::iRFP670 virus was generated by cloning AAV vector serotype 2 ITRs with a rat Synapsin promoter (a gift from Hirai and colleagues[66]), a iRFP670 coding sequence (a gift from Shcherbakova and Verkhusha[67], Addgene plasmid 45457), WPRE, and human growth hormone polyA sequence. Viral production and purification was performed according to McClure et al.[68] with minor modifications. Briefly, 15 × 15 cm dishes of HEK293T cells were transfected with the pAAV-rSyn::iRFP670 plasmid, together with pAAV1, pAAV2, and the helper plasmid pFdelta6 using PEI (Sigma, 408727). Seventy-two hours after transfection, the cells were collected and lysed according to the protocol and the virus containing cell lysate was loaded onto HiTrap Heparin columns (GE Biosciences 17-0406-01). After a series of washes, the virus was eluted from the heparin column to a final concentration of 450 mM NaCl. Finally, the virus was concentrated using Amicon Ultra-15 centrifugal filter units (Millipore UFC910024) and semi-quantitative titering of viral particles was executed by Coomassie staining versus standards. CA3 neurons of hippocampal slices were infected with AAV-rSyn::iRFP670 the day following the preparation of the culture, by a local injection of 0.1 μL of virus in the pyramidal layer of CA3 using a glass capillary under a stereomicroscope. Expression of iRFP was observed in CA3 neurons exclusively as early as 5 days post infection, and expression in CA3-CA1 Schaffer collaterals reached satisfactory expression level 10 days after infection.

**Analysis of microglia–presynaptic compartment interactions**. Contacts between microglia and boutons were detected by scouring iRFP + axons in three dimensions for the entire duration of the imaging session. Elimination events were defined as clear engulfment of iRFP + material by microglia from an iRFP + bouton. Latency before engulfment was measured as the time of contact between microglia and the bouton before any iRFP + material was seen internalized. Inclusions were defined as iRFP + structures within microglia for which the origin could not be defined, as they were present from the beginning of the imaging session.

**Analysis of microglia–postsynaptic compartment interactions**. Contacts between microglia and the postsynaptic compartment were detected by scouring dendritic shafts of GFP + neurons in three-dimensions for the entire duration of the imaging session. All z-planes containing the contacted spine over time were axially projected. GFP intensity was measured in the dendritic shaft and normalized across all the datasets. GFP signal noise and residual microglial YFP signal were measured in microglial processes, and thresholded out to the measured value + 40%. For all spines that were seen to be contacted at least once by microglia, we measured the spine length, the size of the head, and the extent of contact between the spine and microglia for each timepoint of the entire imaging session. The length of the spine was measured from the dendritic shaft to the tip of the spine head, the change in head size was measured as variation in GFP signal at the spine head, and the extent of contact was measured as the percentage of GFP signal at the spine head colocalizing with tdTomato signal. We then performed cross-correlation analysis over time of the extent of contact, with the spine length variation (correlation with the head size not shown) using bootstrap resampling on MATLAB.

**Statistical analysis**. All data are represented as mean ± SEM. To determine statistical significance, data distribution was first tested for variance and normality,

and the corresponding t-test (parametric or non-parametric) was performed using the Graphpad Prism software. For multiple group analysis (motility analysis over time) two-way ANOVA was performed using Graphpad Prism software. Cross-correlation analysis was made using MATLAB software.

**Data availability**. All relevant data are available from the authors.

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

## Acknowledgements

We thank Hiroki Asari for his help with the cross-correlation analysis using MATLAB, Andreas Buness for vectorial correlation on MATLAB, Tom Boissonnet for generating the CLEM animation with Blender, Cristian Tischer for improvement of Image J drift-correction's script, and Senthilkumar Deivasigamani for careful reading of the manuscript and critical input. Funding was provided by EMBL (C.T.G. and L.W.), ERC Advanced Grant COREFEAR (C.T.G), and the People Programme of the European Union's Seventh Framework Programme FP7/2007-2013/ under REA grant agreement number 327409 (U.N.).

## Author contributions

L.W. and C.T.G. designed the experiments. L.W. performed the experiments and analyzed the data, except as follows. G.d.B. performed the microglia motility analysis and spine head filopodia characterization. P.M. and N.S. performed FIB-SEM acquisition under the supervision of Y.S. and G.B. U.N. and A.V. performed the analysis of microglia phagocytic capacity. M.E. performed microglia-presynaptic interactions analysis. A.R. produced the AAV-*rSyn::*iRFP virus. A.S. performed pilot FIB-SEM acquisition. L.W. and C.T.G. wrote the paper.

## Additional information

**Competing interests:** The authors declare no competing interests.

