## [Peer Review File(PDF 491 kb) · Nature Communications]

Reviewers' comments:

Reviewer #1 (Remarks to the Author):

The authors present an impressive experimental effort linking fluorescence microscopy data on microglia-synapse interaction in mice with structural electron microscopy data from exactly the same region of interest. This is a very complex and tedious experimental procedure requiring many intermediate steps such as UV branding the tissue and X-ray micro-tomography to monitor the ROI during preparation for electron microscopy and align both the data sets. The Schwab group has been pioneering this approach and this is another example where they are leading the way. Clearly, with such a laborious workflow, the number of correlative events analyzed is low, which is reflected in the fact that the statistics in Fig 2f is built from 8 and 5 observed apposition and encapsulation events respectively. This is however not obvious from the figure as it is presented percentage-wise and I think the actual N=8 and N=5 should be mentioned on top of both bars. With diffraction-limited confocal microscopy, one may expect a considerable number of fluorescence-classified encapsulation events to actually not be physically encapsulated when observed with 10nm resolution EM, which is obviously shown in the figure, but the percentages are only roughly indicative. While I do not find myself sufficiently experienced to comment on the neurobiological relevance of the work, I think the experimental work in itself and the underpinning of fluorescence classification in neuronal development with EM data is a hallmark example in itself that is worth publishing in Nature Communications. Both in terms of experimental workflow and data analysis, it sets the stage for 3D correlative experiments in developmental neurobiology. I only have one other issue that I feel should be addressed before the manuscript can be accepted: it is not clear from the descriptions whether the data in Fig 2 (events mentioned in 2f) and also Fig 3 originate from multiple tissue blocks or from the same tissue block. In the methods it is mentioned multiple mice were used, but does the final data originate from one ROI, multiple ROI, or multiple ROI in multiple mice. More importantly, is all this data from the same neuron or not? This is not clear, and I think data should preferably originate from at least two mice, but at the minimum from two different neurons in the same mouse.

Reviewer #2 (Remarks to the Author):

In this manuscript, Weinhard et al demonstrate that microglia trogocytose presynaptic axons and boutons and induce postsynaptic filopodia formation. Using mouse lines Cx3cr1-Cre and tdTomato-loxP mice to identify microglia and Thy1-EGFP mice to label dendritic spines, along with immunofluorescence, correlative light-electron microscopy (CLEM), in vivo light sheet imaging of organotypic hippocampal slice cultures with an AAV-Synapsin iRFP marker, the authors find no evidence of spine phagocytosis by microglia. Microglia phagocytosis of presynaptic compartments did not depend on the complement pathway, as microglial trogocytosis was unaffected in Cr3-null animals. Microglia may promote synaptic signaling efficiency by the formation of multiple synaptic boutons (MSBs).

The genetic models are convincing because they clearly identify microglia and their contacts with pre- or postsynaptic structures. Confocal imaging with CLEM yields compelling data to support the notion that microglia do not engulf EGFP-positive postsynaptic spines or filopodia. Moreover, live imaging of hippocampal slice cultures is a powerful tool to examine presynaptic trogocytosis by microglia as well as microglia-mediated filopodia formation.

As noted by the authors, this research area has been active over the past five years and many of the processes touched on here have been addressed by others. Nonetheless, the data and discussion in this report extend our concept of microglial pruning of synapses in two ways: first but demonstrating trogocytosis as contrasted with phagocytosis. Removal of large membrane elements from neurons (as proposed by the hypothesis of synaptic element phagocytosis) would potentially compromise neuronal viability. Trogocytosis by definition involves a non-harmful interaction between cells (at least when one is not considering protozoal parasites) and raises less concern about how microglia might interact with neurons during circuit refinement. Second, the demonstration of post-synaptic filopodia formation, potentially leading to MSBs, raises the intriguing possibility that microglia could promote plasticity by a novel mechanism.

Modifications to the manuscript would improve the paper's clarity and depth:

1. "Notably, inhibitory synapses appear to be unaffected by disruptions of neuron-microglia signaling¹¹".

Comment: This assertion is not entirely correct: J Cell Biol. 2017 Sep 4;216(9):2979-2989. doi: 10.1083/jcb.201607048. Epub 2017 Jul 17. Microglia control the glycinergic but not the GABAergic synapses via prostaglandin E2 in the spinal cord. Cantaut-Belarif Y1, Antri M2, Pizzarelli R1, Colasse S1, Vaccari I1, Soares S3, Renner M1, Dallel R2, Triller A4, Bessis A5.

2. "...first direct evidence for the phagocytic elimination..."

Comment: You're drawing a distinction between phagocytosis and trogocytosis in this report. The cell-biology and receptor utilization in the two processes are overlapping but at least partially distinct. It will be important for clarity's sake to be entirely precise and consistent about terminology in your manuscript. You state near the end of the paper that the endocytic machinery involved in phagocytosis and trogocytosis are similar and partially overlap. It will be useful to draw the particle size distinctions and clarify why the differentiation matters, in the paper's introduction.

3. "For example, immunoreactivity for postsynaptic density 95 (PSD95) protein was found to be localized inside microglia by confocal, super-resolution, and electron microscopy."

Comment: Two points to consider: First, trogocytosis often involves exchange of membrane components between cells. Non-phagocytic but contact dependent exchange of membrane components also occurs (Physiology (Bethesda). 2013 Nov;28(6):414-22. doi: 10.1152/physiol.00032.2013). Given the extensive interaction between microglia and spines, exchange of membrane proteins is plausible. Second, PSD95 may not be the best

example here: note that PSD95/dlg4 is also expressed in microglia (http://web.stanford.edu/group/barres_lab/cgi-bin/igv/cgi_2.py?In_ame=Dlg4).

4. Figure 1b-d

Comment: The intact spines shown in this figure resemble long-thin (1b) or mushroom (1d) spine morphology. Do the authors feel that these morphologies are more likely to give rise to artefactual reports of spine engulfment?

5. "However, contrary to previous assumptions, we found no evidence for the phagocytosis of entire synapses. Instead we observed microglia trogocytosis – or nibbling – of synaptic structures. Importantly, microglia trogocytosis was restricted to presynaptic boutons and axons, with no evidence for engulfment of postsynaptic elements."

Comment: Given that much of the data, although high-quality and abundant, are negative, it's important to point out the limitations of the present study, including the fact that only one CNS region, in one species, at one time point was examined.

6. The authors should clarify the putative mechanism shown in Figure 3f by presenting the EM data next to a schematic, as the images have not been conclusively demonstrated to be a timecourse of the same synapse undergoing the listed steps.

7. It is not entirely convincing that Figure 3c demonstrates autophagocytosis (in which a microglial cell is engulfing itself), nor is it clear what the significance might be for synapses. Additional discussion of its functional impact in their current model should be provided.

Reviewer #3 (Remarks to the Author):

Overall Evaluation:

The role of microglia trogocytosis (nibbling) in synapse remodeling is an important current topic in neuroscience. Hence, further understanding about what this process entails would be most welcome. There have been many outstanding papers regarding the roles of microglia in synapse remodeling during neurodegeneration as well as normal development. These studies have resulted in a consensus that microglia participate actively in synapse pruning. Prior studies used stimulated STED and immunoelectron microscopy to demonstrate that both postsynaptic components (e.g. PSD95), and presynaptic components (e.g. SNAP95), are engulfed by, or at least partially engulfed by microglia. The present paper raised the question as to whether both pre- and postsynaptic components were indeed phagocytosed by microglia.

The combination of correlative light and electron microscopy (CLEM) , using confocal, light-sheet and FIB-SEM (focused ion beam – serial electron microscopy), as well as microCT are a strength of this paper. The goal was to address especially the controversial question of whether both presynaptic and postsynaptic components were trogocytosed. The authors

arrive at the conclusion that only presynaptic components are trogocytosed, by microglia. This conclusion relies heavily on the quality of the FIB-SEM images to distinguish presynaptic components from postsynaptic components. There are many interesting STED observations about the interactions between microglia and neurons, including KO mouse control experiments. The STED images are excellent and compelling.

Unfortunately, the main observations from FIB-SEM images that are needed to arrive at the novel conclusion regarding pre, but not postsynaptic trogocytosis, are not convincing. The resolution of the FIB-SEM images is simply not sufficient to convince this reader that even the best examples of the inclusions found through FIB-SEM arose specifically from axons. Dendrites, astroglia, growth cones, and cell bodies also contain accumulations of vesicles, especially at young ages. The few available example images have just a few vesicles inside, and hence by themselves do not unequivocally indicate axons, presynaptic boutons or other objects. Furthermore, the clear majority of the inclusions observed in FIB-SEM were 'undetermined' as to their origin. Hence, there is simply insufficient evidence to rule out postsynaptic trogocytosis as a major contributor, as other publications have shown or suggested (e.g. Paolicelli, R. C. et al. Synaptic pruning by microglia is necessary for normal brain development. *Science* 333, 1456–8 (2011)).

Other issues that would improve the paper if properly addressed include:

Need clarifications/better definitions for the following:

- "contact" between microglia and spines in FIB-SEM data set (Fig 2)
- "putative presynaptic vesicles" (Fig 3a)
- "filopodia" and "spine head filopodia" – for Figs 6 and 7. Some of them look short. Any relevance to spinules? How do the spine head filopodia relate to spine head protrusions described by other scientists (e.g. RA McKinney).
- "relocation of the spine head" (Fig 7a)
- How is filopodial (or SHP) outgrowth distinguished from active stretching by the microglia. The authors repeatedly discuss 'stretching' but the videos and images presented really could equally be discussed as outgrowth – which might be initiated from the postsynaptic side.

The introduction stresses as a potential role of microglia that they may participate in brain homeostasis – but this manuscript does not address this topic. Suggest that the focus be on the primary topics of the paper, just mentioning homeostasis as one among many functions.

The last sentence of Intro – the authors argue that they have provided "direct evidence for the phagocytic elimination of synapses by microglia". However, at best, they show that microglia nibble on axons. There is no clear evidence that synapses are eliminated. Synapses are not actually counted and the live imaging does not illustrate synapse elimination, just partial remodeling, at best. This concern applies to the Discussion, as well.

Fig. 2F – I do not understand what 'confirmed' means in the "encapsulation" components. Does it mean confirmed encapsulation – if so, then this graph lends further doubt to the main conclusion, namely that some spines might in fact be trogocytosed.

Fig 3d: The authors indicate a very small volume was investigated (just 50 μm^2). Perhaps greater confidence in the observations would be achieved by sampling more volumes in the FIB-SEM. Also, does presynaptic = confirmed boutons, and if so, how were they confirmed? "Importantly, microglia at the imaging site did not show any detectable morphological changes following viral infection as they were imaged two weeks later and 500 μm distant from the infection site." – This statement needs quantitative backing, perhaps as a supplemental Figure?

Fig 4 – "a stack-projection" – how many optical slices over what z-distance? In (b), yellow is hard to see.

Data for Fig 5 needs a positive control to show that the transgenic manipulation (knocking out CR3) affects other known CR3-dependent functions of microglia.

Fig 7e-j shows convergence of numerous filopodia onto a single microglial process – how common is this phenomenon? If this is a rare phenomenon, then the authors should be careful about any conclusions drawn. In 7h, white PSD is hard to see against yellow.

Last paragraph of Results: ... "one appeared to initiate a synapse, resulting in the formation of a multiple synaptic bouton" – What exactly do the authors mean by "initiate a synapse".

Supplementary Figure 1: How were these images obtained? How was fluorescence intensity measured? Do all microglia express Iba1 and Cx3cr1?

Tissue prep for light-sheet imaging involves some drastic temperature changes to mount cultured tissue on agarose. It would be good to discuss how these temperature changes could affect neuron/microglia structure.

For light-sheet imaging, a 585nm long-pass filter was used for both tdTomato and iRFP – what is the degree of overlap between emissions from these fluorophores at wavelength longer than 585nm?

Incomplete Citation list:

Missing from the citations are several Key articles regarding spine head protrusions and filopodia; as well as prior evidence for postsynaptic trogocytosis – by structures other than microglia, such as axons and astroglia, that could strengthen their arguments including:

1. PIP_3 regulates spinule formation in dendritic spines during structural long-term potentiation.

Ueda Y, Hayashi Y. *J Neurosci*. 2013 Jul 3;33(27):11040-7.

2. Rewiring neuronal microcircuits of the brain via spine head protrusions--a role for synaptopodin and intracellular calcium stores. Verbich D, Becker D, Vlachos A, Mundel P, Deller T, McKinney RA. *Acta Neuropathol Commun*. 2016 Apr 22;4:38. doi:

10.1186/s40478-016-0311-x.

3. Prolonged ampkine exposure prunes dendritic spines and increases presynaptic release probability for enhanced long-term potentiation in the hippocampus. Chang PK, Prenosil GA, Verbich D, Gill R, McKinney RA. *Eur J Neurosci*. 2014 Sep;40(5):2766-76. doi: 10.1111/ejn.12638. Epub 2014 Jun 13.
4. Glial glutamate transport modulates dendritic spine head protrusions in the hippocampus. Verbich D, Prenosil GA, Chang PK, Murai KK, McKinney RA. *Glia*. 2012 Jul;60(7):1067-77. doi: 10.1002/glia.22335. Epub 2012 Apr 4.
5. Rapid and reversible formation of spine head filopodia in response to muscarinic receptor activation in CA1 pyramidal cells. Schätzle P, Ster J, Verbich D, McKinney RA, Gerber U, Sonderegger P, Mateos JM. *J Physiol*. 2011 Sep 1;589(17):4353-64. doi: 10.1113/jphysiol.2010.204446. Epub 2011 Jul 18.
6. Dendritic spine morphology determines membrane-associated protein exchange between dendritic shafts and spine heads. Hugel S, Abegg M, de Paola V, Caroni P, Gähwiler BH, McKinney RA. *Cereb Cortex*. 2009 Mar;19(3):697-702. doi: 10.1093/cercor/bhn118. Epub 2008 Jul 24.
7. Richards DA, Mateos JM, Hugel S, de Paola V, Caroni P, Gähwiler BH, McKinney RA. *Proc Natl Acad Sci U S A*. 2005 Apr 26;102(17):6166-71. Epub 2005 Apr 14.
8. Trans-endocytosis via spinules in adult rat hippocampus. Spacek J, Harris KM. *J Neurosci*. 2004 Apr 28;24(17):4233-41.

Some of the Citations are incomplete, missing journal names for example:

Hong, S. et al. Complement and Microglia Mediate Early Synapse Loss in Alzheimer Mouse Models. 352, 712–716 (2016).

Rebuttal: NCOMMS-17-22931

All modifications to the manuscript text are highlighted in red font.

Reviewers' comments:

Reviewer #1 (Remarks to the Author):

The authors present an impressive experimental effort linking fluorescence microscopy data on microglia-synapse interaction in mice with structural electron microscopy data from exactly the same region of interest. This is a very complex and tedious experimental procedure requiring many intermediate steps such as UV branding the tissue and X-ray micro-tomography to monitor the ROI during preparation for electron microscopy and align both the data sets. The Schwab group has been pioneering this approach and this is another example where they are leading the way. Clearly, with such a laborious workflow, the number of correlative events analyzed is low, which is reflected in the fact that the statistics in Fig 2f is built from 8 and 5 observed apposition and encapsulation events respectively. This is however not obvious from the figure as it is presented percentage-wise and I think the actual N=8 and N=5 should be mentioned on top of both bars. With diffraction-limited confocal microscopy, one may expect a considerable number of fluorescence-classified encapsulation events to actually not be physically encapsulated when observed with 10nm resolution EM, which is obviously shown in the figure, but the percentages are only roughly indicative. While I do not find myself sufficiently experienced to comment on the neurobiological relevance of the work, I think the experimental work in itself and the underpinning of fluorescence classification in neuronal development with EM data is a hallmark example in itself that is worth publishing in Nature Communications. Both in terms of experimental workflow and data analysis, it sets the stage for 3D correlative experiments in developmental neurobiology. I only have one other issue that I feel should be addressed before the manuscript can be accepted: it is not clear from the descriptions whether the data in Fig 2 (events mentioned in 2f) and also Fig 3 originate from multiple tissue blocks or from the same tissue block. In the methods it is mentioned multiple mice were used, but does the final data originate from one ROI, multiple ROI, or multiple ROI in multiple mice. More importantly, is all this data from the same neuron or not? This is not clear, and I think data should preferably originate from at least two mice, but at the minimum from two different neurons in the same mouse.

Author Response: *We agree with the reviewer that it is not entirely surprising that a significant fraction of encapsulation events identified by confocal microscopy would not be confirmed by EM, but were nevertheless surprised by the poor correlation. We have highlighted this finding as important because the field has routinely used confocal fluorescence colocalization microscopy to quantify microglia-neuron interaction events and our data suggest that these data are easily misinterpreted. Following the reviewer's suggestion we have now added the absolute data values above the graphs. Apposition/encapsulation events in **Fig. 2** derive from 4 ROIs containing 4 neurons from a total of 2 animals. Microglial content analysis in **Fig. 3**, on the other hand, derives from 7 ROIs containing end-processes from 8 microglia from a total of 4 animals. These numbers are now explicitly mentioned in the Materials & Methods section (page 23, line 496).*

Reviewer #2 (Remarks to the Author):

In this manuscript, Weinhard et al demonstrate that microglia trogocytose presynaptic axons and boutons and induce postsynaptic filopodia formation. Using mouse lines Cx3cr1-Cre and tdTomato-loxP mice to identify microglia and Thy1-EGFP mice to label dendritic spines, along with

immunofluorescence, correlative light-electron microscopy (CLEM), in vivo lightsheet imaging of organotypic hippocampal slice cultures with an AAV-Synapsin iRFP marker, the authors find no evidence of spine phagocytosis by microglia. Microglia phagocytosis of presynaptic compartments did not depend on the complement pathway, as microglial trogocytosis was unaffected in Cr3-null animals. Microglia may promote synaptic signaling efficiency by the formation of multiple synaptic boutons (MSBs).

The genetic models are convincing because they clearly identify microglia and their contacts with pre- or postsynaptic structures. Confocal imaging with CLEM yields compelling data to support the notion that microglia do not engulf EGFP-positive postsynaptic spines or filopodia. Moreover, live imaging of hippocampal slice cultures is a powerful tool to examine presynaptic trogocytosis by microglia as well as microglia-mediated filopodia formation.

As noted by the authors, this research area has been active over the past five years and many of the processes touched on here have been addressed by others. Nonetheless, the data and discussion in this report extend our concept of microglial pruning of synapses in two ways: first but demonstrating trogocytosis as contrasted with phagocytosis. Removal of large membrane elements from neurons (as proposed by the hypothesis of synaptic element phagocytosis) would potentially compromise neuronal viability. Trogocytosis by definition involves a non-harmful interaction between cells (at least when one is not considering protozoal parasites) and raises less concern about how microglia might interact with neurons during circuit refinement. Second, the demonstration of post-synaptic filopodia formation, potentially leading to MSBs, raises the intriguing possibility that microglia could promote plasticity by a novel mechanism.

Modifications to the manuscript would improve the paper's clarity and depth:

1. "Notably, inhibitory synapses appear to be unaffected by disruptions of neuron-microglia signaling¹¹".

Comment: This assertion is not entirely correct: J Cell Biol. 2017 Sep 4;216(9):2979-2989. doi: 10.1083/jcb.201607048. Epub 2017 Jul 17. Microglia control the glycinergic but not the GABAergic synapses via prostaglandin E2 in the spinal cord. Cantaut-Belarif Y1, Antri M2, Pizzarelli R1, Colasse S1, Vaccari I1, Soares S3, Renner M1, Dallel R2, Triller A4, Bessis A5.

Author Response: *We have changed the wording to be more precise and include the citation as follows: "Notably, inhibitory synapses **in hippocampus** appear to be unaffected by disruptions of neuron-microglia signalling (but see **Cantaut-Belarif et al., 2017**)." (page 3, line 63).*

2. "...first direct evidence for the phagocytic elimination..."

Comment: You're drawing a distinction between phagocytosis and trogocytosis in this report. The cell-biology and receptor utilization in the two processes are overlapping but at least partially distinct. It will be important for clarity's sake to be entirely precise and consistent about terminology in your manuscript. You state near the end of the paper that the endocytic machinery involved in phagocytosis and trogocytosis are similar and partially overlap. It will be useful to draw the particle size distinctions and clarify why the differentiation matters, in the paper's introduction.

Author Response: *Good points – we have now refined our terminology and added a description clarifying the distinction between phagocytosis and trogocytosis in the introduction: "Trogocytosis has been described in the immune system as a non-apoptotic mechanism for the rapid capture of membrane components and differs from phagocytosis which involves the engulfment of larger cellular structures (>1µm)¹⁷⁻¹⁹." (page 4, line 83).*

3. “For example, immunoreactivity for postsynaptic density 95 (PSD95) protein was found to be localized inside microglia by confocal, super-resolution, and electron microscopy.”

Comment: Two points to consider: First, trogocytosis often involves exchange of membrane components between cells. **Non-phagocytic but contact dependent exchange of membrane components also occurs (Physiology (Bethesda). 2013 Nov;28(6):414-22. doi: 10.1152/physiol.00032.2013)**. Given the extensive interaction between microglia and spines, exchange of membrane proteins is plausible. Second, PSD95 may not be the best example here: note that PSD95/dlg4 is also expressed in microglia (http://web.stanford.edu/group/barres_lab/cgi-bin/igv.cgi_2.py?lname=Dlg4).

Author Response: *We now reference the mentioned citations as possible mechanisms for PSD95 protein localization in microglia: “...it should be noted that PSD95 has been found to be expressed by microglia⁴⁷ (but see ref.⁴⁸), and that immunocolocalization of synaptic proteins with microglia might derive from non-phagocytic membrane exchange⁴⁹.” (page 19, line 375).*

4. Figure 1b-d

Comment: The intact spines shown in this figure resemble long-thin (1b) or mushroom (1d) spine morphology. Do the authors feel that these morphologies are more likely to give rise to artefactual reports of spine engulfment?

Author Response: *In fact, systematic analysis of our confocal data shows that thin spines are more likely to appear encapsulated compared to mushroom spines (data not shown). We agree with the reviewer that this could be the result of the fact that thin objects might be more likely to appear entirely surrounded by microglia fluorescence than large ones.*

5. “However, contrary to previous assumptions, we found no evidence for the phagocytosis of entire synapses. Instead we observed microglia trogocytosis – or nibbling – of synaptic structures. Importantly, microglia trogocytosis was restricted to presynaptic boutons and axons, with no evidence for engulfment of postsynaptic elements.”

Comment: Given that much of the data, although high-quality and abundant, are negative, it's important to point out the limitations of the present study, including the fact that only one CNS region, in one species, at one time point was examined.

Author Response: *We have added a sentence in the discussion to stress this point: “...it remains possible that elimination of postsynaptic material by microglia might occur in other brain regions, at other developmental stages, or alternatively at such a low frequency that it went unnoticed in our experiments...” (page 19, line 377).*

6. The authors should clarify the putative mechanism shown in Figure 3f by presenting the EM data next to a schematic, as the images have not been conclusively demonstrated to be a timecourse of the same synapse undergoing the listed steps.

Author Response: *A schematic has been added above each image column and the figure legend indicate that they represent a putative series of events, rather than an actual time course.*

7. It is not entirely convincing that Figure 3c demonstrates autophagocytosis (in which a microglial cell is engulfing itself), nor is it clear what the significance might be for synapses. Additional discussion of its functional impact in their current model should be provided.

Author Response: *As an incidental finding to our investigation of microglia-neuron interactions, we frequently found microglial processes to be enwrapping portions of the same microglia (Fig. 3). These interactions were so intimate, frequently involving the pinching of plasma membrane, that we interpret them as intermediates in autophagocytosis. We have now added additional examples of such interactions to the manuscript to better illustrate them (Supplementary Fig. 2). However, we agree that strictly speaking we cannot rule out that these are only transient intimate interactions that resolve without phagocytosis. We have therefore renamed this phenomenon “self-engulfment” to make it match more accurately our observation (page 8, line 162). It is difficult to speculate what the functional impact of self-engulfment might be, but given its unexpected abundance we felt it was worth noting. Nevertheless, in accordance with it not being the central thesis of the manuscript, we have now moved the image sequences showing microglial self-engulfment to the supplementary data section (Supplementary Fig. 2).*

Reviewer #3 (Remarks to the Author):

Overall Evaluation:

The role of microglia trogocytosis (nibbling) in synapse remodeling is an important current topic in neuroscience. Hence, further understanding about what this process entails would be most welcome. There have been many outstanding papers regarding the roles of microglia in synapse remodeling during neurodegeneration as well as normal development. These studies have resulted in a consensus that microglia participate actively in synapse pruning. Prior studies used stimulated STED and immunoelectron microscopy to demonstrate that both postsynaptic components (e.g. PSD95), and presynaptic components (e.g. SNAP95), are engulfed by, or at least partially engulfed by microglia. The present paper raised the question as to whether both pre- and postsynaptic components were indeed phagocytosed by microglia.

The combination of correlative light and electron microscopy (CLEM), using confocal, light-sheet and FIB-SEM (focused ion beam – serial electron microscopy), as well as microCT are a strength of this paper. The goal was to address especially the controversial question of whether both presynaptic and postsynaptic components were trogocytosed. The authors arrive at the conclusion that only presynaptic components are trogocytosed, by microglia. This conclusion relies heavily on the quality of the FIB-SEM images to distinguish presynaptic components from postsynaptic components. There are many interesting STED observations about the interactions between microglia and neurons, including KO mouse control experiments. The STED images are excellent and compelling.

Unfortunately, the main observations from FIB-SEM images that are needed to arrive at the novel conclusion regarding pre, but not postsynaptic trogocytosis, are not convincing. The resolution of the FIB-SEM images is simply not sufficient to convince this reader that even the best examples of the inclusions found through FIB-SEM arose specifically from axons. Dendrites, astroglia, growth cones, and cell bodies also contain accumulations of vesicles, especially at young ages. The few available example images have just a few vesicles inside, and hence by themselves do not unequivocally indicate axons, presynaptic boutons or other objects. Furthermore, the clear majority of the inclusions observed in FIB-SEM were ‘undetermined’ as to their origin. Hence, there is simply insufficient evidence to rule out postsynaptic trogocytosis as a major contributor, as other publications have shown or suggested (e.g. Paolicelli, R. C. et al. Synaptic pruning by microglia is necessary for normal brain development. *Science* 333, 1456–8 (2011)).

Author Response: *We agree with the reviewer that conclusions based solely on the identity of complete inclusions found in microglia by FIB-SEM would be weak. However, our conclusions for*

primarily presynaptic trogocytosis are based on a wider set of observations, perhaps not sufficiently elaborated in the manuscript: 1) absence of phagocytosis/trogocytosis of postsynaptic material by microglia as assessed by confocal microscopy in fixed brain sections (**Fig. 1-2**), 2) frequent and intimate partial inclusion events (“pinched” material) of exclusively presynaptic boutons and axons as assessed by 3D volume FIB-SEM (**Fig. 3**), and 3) intake of presynaptic (**Fig. 4**), but not postsynaptic material by microglia as assessed by time-lapse fluorescence imaging (**Fig. 6**). We feel that the combination of these three data allow us to rule out a major role for microglia elimination of postsynaptic/dendritic material in the developing hippocampus and we have now clarified this claim in the discussion (page 18, line 357). Of course, it remains possible that postsynaptic trogocytosis by microglia occurs under conditions other than those examined in our preparations and we have added a cautionary phrase to this effect in the discussion: “...although it remains possible that elimination of postsynaptic material by microglia might occur in other brain regions, at other developmental stages, or alternatively at such a low frequency that it went unnoticed in our experiments” (page 19, line 377). Finally, it’s important to point out that we believe that the major contribution of our work is to 1) provide the missing direct evidence for microglia engulfment of neuronal material, 2) caution the field about the dangers inherent in estimating engulfment from fluorescence colocalization studies in fixed tissue, and 3) identify the induction of spine head filopodia as a common microglia-synapse interaction. Obviously, ruling out a major role for microglia phagocytosis of postsynaptic structures is also a major finding, but given that it is based on negative data it is inherently more difficult to completely exclude – as the review points out. We hope that the changes we have made in response to the reviewers have better presented our critical findings.

Other issues that would improve the paper if properly addressed include:

Need clarifications/better definitions for the following:

- “contact” between microglia and spines in FIB-SEM data set (Fig 2) :

Author response: “Contact” refers to a juxtaposition or apposition of microglia and spine membranes. This term is now defined in the text (page 6, line 145).

- “putative presynaptic vesicles” (Fig 3a)

Author response: “Putative presynaptic vesicles” are vesicles assumed to be presynaptic due to their 40 nm diameter size (page 8, line 157).

- “filopodia” and “spine head filopodia” – for Figs 6 and 7. Some of them look short. Any relevance to spinules? How do the spine head filopodia relate to spine head protrusions described by other scientists (e.g. RA McKinney)

Author response: The spinules described by Spacek et al. (2004) are typically short (0.2-0.5 μm) and engulfed by material from a single cell (axon or astrocyte). In our FIB-SEM reconstructions, spine head filopodia are always found in the extracellular space between cells and vary in length from 0.4 to 3.1 μm with an average of 1.5 μm . Our spine head filopodia closely resemble protrusions observed in hippocampal slices by Richards et al. (2005), and more recently in the olfactory bulb by Provencher-Breton et al. (2016). It is likely that spine head protrusions and spine head filopodia are functionally similar structures and we now discuss the literature on spine head protrusions in the text (page 19, line 392).

- “relocation of the spine head” (Fig 7a)

Author response: “Relocation of the spine head” refers to the displacement of the spine head from its original site to the tip of the spine head filopodia (page 14, line 277).

- How is filopodial (or SHP) outgrowth distinguished from active stretching by the microglia. The authors repeatedly discuss ‘stretching’ but the videos and images presented really could equally be discussed as outgrowth – which might be initiated from the postsynaptic side.

Author response: We absolutely agree with the reviewer that we cannot draw any conclusions about whether “stretching” events are caused by microglia-dependent application of tension or are simply a microglia-associated growth response of the spine head. It is possible that microglia trigger the formation of spine head filopodia by releasing BDNF or glutamate as both of these factors were shown to promote spine head filopodia (Richards et al., 2005; Breton-provencher et al., 2016). Alternatively, microglia could indirectly induce spine head filopodia formation by “loosening up” the local extracellular environment. These points are now mentioned in the Discussion (page 20, line 402).

The introduction stresses as a potential role of microglia that they may participate in brain homeostasis – but this manuscript does not address this topic. Suggest that the focus be on the primary topics of the paper, just mentioning homeostasis as one among many functions.

Author response: We have reworded the Introduction to be more precise: “The recent discovery that microglia are also highly motile in the uninjured brain^{2,3}, continuously extending and retracting processes through the extracellular space, suggests that they may monitor and contribute to synaptic maturation and function.” (page 3, line 46).

The last sentence of Intro – the authors argue that they have provided “direct evidence for the phagocytic elimination of synapses by microglia”. However, at best, they show that microglia nibble on axons. There is no clear evidence that synapses are eliminated. Synapses are not actually counted and the live imaging does not illustrate synapse elimination, just partial remodeling, at best. This concern applies to the Discussion, as well.

Author response: We completely agree with the reviewer and apologize if it appeared we were over-interpreting our findings. We have now added sentences to stress this point in the Introduction – “Our findings provide the first direct evidence for the elimination of **synaptic material** by microglia in living brain tissue” (page 4, line 90) and Discussion – “It should also be noted that the elimination of presynaptic material does not necessarily imply elimination of functional synaptic structures...” (page 18, line 348).

Fig. 2F – I do not understand what ‘confirmed’ means in the “encapsulation” components. Does it mean confirmed encapsulation – if so, then this graph lends further doubt to the main conclusion, namely that some spines might in fact be trogocytosed.

Author response: We use the term “confirmed” to indicate that both light and electron microscopy approaches are in agreement about the extent of microglia-synapse contact (we use the term “apposition” to refer to minimal contact events, “encapsulation” to refer to major contact events where the spine appears partially engulfed but is still attached to the dendrite, and “inclusion” to refer to complete engulfment events). In light microscopy, to be considered encapsulated a spine had to have at least 70% of its surface in contact with microglia. Considering the limitations of light microscopy under the tissue conditions used for CLEM, we lowered this threshold to 50%

when analysing those contacts by FIB-SEM. Of the 5 encapsulated spines studied by CLEM, one turned out by FIB-SEM not to be in contact with microglia, 3 turned out to be simple appositions (<50% of neuronal spine in contact with microglia), and only one turned out to be contacted at 50% of its surface and therefore reached the threshold to be classified as “confirmed”, with no sign of phagocytosis/trogocytosis. Thus, these data show that the extent of interaction observed in light microscopy is not confirmed by electron microscopy, and argue that fluorescence-based approaches widely used in the field can be misleading as methods to test engulfment of synaptic material by microglia. Finally, it should be noted that the high surface/volume ratio of the postsynaptic compartment means that microglia are likely to contact dendritic spines frequently by chance.

Fig 3d: The authors indicate a very small volume was investigated (just 50 μm^2). Perhaps greater confidence in the observations would be achieved by sampling more volumes in the FIB-SEM. Also, does presynaptic = confirmed boutons, and if so, how were they confirmed?

Author response: For the analysis of microglial content we imaged 7 tissue blocks (>2000 image sections/block for about 17,000 μm^3 analysed) from which we 3D-reconstructed all the microglial processes present (8 in total) as identified by correlation with confocal images. Since these were thin microglia end-processes, the volume is relatively small (56 μm^3) compared to the surface analysed (560 μm^2) which is probably a better measure of the extent of the analysis. In **Fig. 3c**, “presynaptic” refers to confirmed presynaptic boutons (we updated the wording in the figure legend). Complete presynaptic bouton inclusions were identified by the presence of 40 nm vesicles typical of presynaptic machinery. Partial presynaptic bouton and axonal material inclusions were identified after 3D reconstruction of the belonging axon, based on the presence of presynaptic machinery: clusters of 40 nm presynaptic vesicles, and apposition on a postsynaptic density. We now better describe how we identified neuronal structures in the Materials and Methods (page 25, line 572).

“Importantly, microglia at the imaging site did not show any detectable morphological changes following viral infection as they were imaged two weeks later and 500 μm distant from the infection site.” – This statement needs quantitative backing, perhaps as a supplemental Figure?

Author response: We have now added data of microglia branching and soma area and shape as a measure of microglia activation/disturbance at the imaging site (**Supplementary Fig. 4**). These data argue against any detectable differences in microglia morphology in CA1 due to viral infection of the slice in CA3. Again, it is important to point out that infection occurred more than 500 μm away from the imaging site and two weeks before imaging, and that no neuronal or microglia cell bodies at the imaging site were infected.

Fig 4 – “a stack-projection” – how many optical slices over what z-distance? In (b), yellow is hard to see.

Author response: The image is a maximal intensity projection of 35 consecutive optical sections of 480 nm each; this information is now indicated in the figure legend. We changed the color to improve visualization.

Data for Fig 5 needs a positive control to show that the transgenic manipulation (knocking out CR3) affects other known CR3-dependent functions of microglia.

Author response: CR3-dependent microglia functions have been well documented in the retinohalamic system (Stevens et al., 2007; Schafer et al., 2012) and more recently in the

hippocampus under pathological conditions (Shi et al., 2015; Hong et al., 2016; Vasek et al., 2016). We used the same mutant allele (Jackson Laboratory stock 003991) as used in these earlier studies and assumed it was not necessary to reproduce these findings as they appear robust and involve methods outside our area of expertise. However, we are currently preparing a manuscript in which we report morphological and electrophysiological defects in postnatal hippocampal neurons of CR3 knockout mice. Given that these deficits are novel CR3-dependent phenotypes, we don't feel that they would serve well as controls in the current work.

Fig 7e-j shows convergence of numerous filopodia onto a single microglial process – how common is this phenomenon? If this is a rare phenomenon, then the authors should be careful about any conclusions drawn. In 7h, white PSD is hard to see against yellow.

Author response: We used volume EM to examine all spines selected by light microscopy for their contact with microglia, and observed that 28% of them (5/18) extended a filopodium toward a nearby microglia process. This includes the case shown in **Fig. 7**, in which we additionally reconstructed all nearby filopodia to reveal an impressive convergence toward the same microglia process. At this point it is not clear how common this type of mass-convergence is as we were not able to carry out the extensive and laborious reconstruction of all spines in the vicinity of microglia that would be required to systematically identify them. However, our time-lapse fluorescent imaging data showed that 39% of spines contacted by microglia (13/31) were associated with spine head filopodia formation and statistical analysis demonstrated that these events significantly correlate in time with microglia contact (**Fig. 6**). Nevertheless, we have added a phrase to indicate the anecdotal nature of the mass-convergence EM finding: "...although caution must be exercised when extrapolating from anecdotal electron microscopy data..." (page 20, line 424).

Last paragraph of Results: ... "one appeared to initiate a synapse, resulting in the formation of a multiple synaptic bouton" – What exactly do the authors mean by "initiate a synapse".

Author response: We used the term "appeared to initiate a synapse" because we observed a small postsynaptic density at the contact between the spine head filopodia and the presynaptic bouton with no clustering of presynaptic vesicles – an architecture suggestive of a newly formed excitatory synapse (page 16, line 298). We have now included an FIB-SEM image sequence of this case in **Supplementary Fig. 5**.

Supplementary Figure 1: How were these images obtained? How was fluorescence intensity measured? Do all microglia express Iba1 and Cx3cr1?

Author response: Iba1 (Ito et al., 1998) and Cx3cr1 (Kim et al., 2011) have been shown to be expressed by both ramified and activated microglia throughout development. In **Supplementary Fig. 1**, the analysis was carried out on Iba1-immunolabeled microglia in fixed brain slices. CD68-immunolabel intensity was measured in Iba1-positive microglia using Imaris software. In addition, we quantified CD68 immunolabeling in genetically tagged microglia to compare CD68 expression patterns. We have reworded our Materials and Methods section to better describe the protocol (page 22, line 463).

Tissue prep for light-sheet imaging involves some drastic temperature changes to mount cultured tissue on agarose. It would be good to discuss how these temperature changes could affect neuron/microglia structure.

Author response: Ex vivo slice culture inevitably involves manipulations that can potentially affect neuron and microglia homeostasis. Mounting of the tissue prior to light sheet imaging required its

immobilization in agarose that involved cooling of the embedded tissue before mounting in the microscope. We limited the cooling to the time needed for the surrounding agarose to stiffen (1 min at 4°C) and it is very unlikely that the tissue temperature dropped to this temperature. Even so, it has been reported that following incubation at 4°C for longer periods of time (>30 min), microglia soma enlarged but recovered normal morphology within 2 hours (Sugama et al., 2011). For this reason, we waited 2 hours for the slice to recover before imaging and carried out control experiments showing that microglia motility (**Supplementary Fig. 3**) was similar to that found by two-photon imaging in vivo (Nimmerjahn et al., 2009.) We are now discussing these aspects in the Material and Methods section (page 26, line 604).

For light-sheet imaging, a 585 nm long-pass filter was used for both tdTomato and iRFP – what is the degree of overlap between emissions from these fluorophores at wavelength longer than 585nm?

Author response: The emission spectra of tdTomato and iRFP overlap significantly and are both efficiently detected in the presence of the 585 nm long-pass filter (see **Fig. R1** below). However, we were able to image them separately because their excitation spectra are distinct, with the 561 nm laser exciting primarily tdTomato (<20% iRFP peak excitation) and the 638 nm laser exciting exclusively iRFP. This allowed us to image iRFP and tdTomato separately by alternating exposure to 561 and 638 nm light sources with a fixed 585 nm long-pass filter detection system. Notably, no detectable iRFP signal was seen during 561 nm illumination, most likely because iRFP is significantly more dim than tdTomato (page 27, line 624).

Incomplete Citation list:

Missing from the citations are several Key articles regarding spine head protrusions and filopodia; as well as prior evidence for postsynaptic trogocytosis – by structures other than microglia, such as axons and astroglia, that could strengthen their arguments including:

1. PIP \square regulates spinule formation in dendritic spines during structural long-term potentiation. Ueda Y, Hayashi Y. J Neurosci. 2013 Jul 3;33(27):11040-7.
2. Rewiring neuronal microcircuits of the brain via spine head protrusions--a role for synaptopodin and intracellular calcium stores. Verbich D, Becker D, Vlachos A, Mundel P, Deller T, McKinney RA. Acta Neuropathol Commun. 2016 Apr 22;4:38. doi: 10.1186/s40478-016-0311-x.
3. Prolonged ampakine exposure prunes dendritic spines and increases presynaptic release probability for enhanced long-term potentiation in the hippocampus. Chang PK, Prenosil GA, Verbich D, Gill R, McKinney RA. Eur J Neurosci. 2014 Sep;40(5):2766-76. doi: 10.1111/ejn.12638. Epub 2014 Jun 13.
4. Glial glutamate transport modulates dendritic spine head protrusions in the hippocampus. Verbich D, Prenosil GA, Chang PK, Murai KK, McKinney RA. Glia. 2012 Jul;60(7):1067-77. doi: 10.1002/glia.22335. Epub 2012 Apr 4.
5. Rapid and reversible formation of spine head filopodia in response to muscarinic receptor activation in CA1 pyramidal cells. Schätzle P, Ster J, Verbich D, McKinney RA, Gerber U, Sonderegger P, Mateos JM. J Physiol. 2011 Sep 1;589(17):4353-64. doi: 10.1113/jphysiol.2010.204446. Epub 2011 Jul 18.
6. Dendritic spine morphology determines membrane-associated protein exchange between dendritic shafts and spine heads. Hugel S, Abegg M, de Paola V, Caroni P, Gähwiler BH, McKinney RA. Cereb Cortex. 2009 Mar;19(3):697-702. doi: 10.1093/cercor/bhn118. Epub 2008 Jul 24.
7. Richards DA, Mateos JM, Hugel S, de Paola V, Caroni P, Gähwiler BH, McKinney RA. Proc Natl Acad Sci U S A. 2005 Apr 26;102(17):6166-71. Epub 2005 Apr 14.
8. Trans-endocytosis via spinules in adult rat hippocampus. Spacek J, Harris KM. J Neurosci. 2004 Apr 28;24(17):4233-41.

Author response: We thank the reviewer for pointing out this very relevant literature. These articles are now discussed (page 19, line 392 and page 20, line 408).

Some of the Citations are incomplete, missing journal names for example:

Hong, S. et al. Complement and Microglia Mediate Early Synapse Loss in Alzheimer Mouse Models. 352, 712–716 (2016).

Author response: These errors have been corrected.

REVIEWERS' COMMENTS:

Reviewer #1 (Remarks to the Author):

The authors have convincingly addressed the comments raised in review. The data requested has been added and is satisfactory. In my opinion also the comments raised by the other reviewers have been dealt with carefully. I think this is an impressive experimental piece of work that should be published in Nature Communications.

Reviewer #3 (Remarks to the Author):

The authors have thoroughly addressed all of the comments and issues raised in the prior reviews.

Rebuttal: NCOMMS-17-22931B

REFEREES' COMMENTS:

Reviewer #1 (Remarks to the Author):

The authors have convincingly addressed the comments raised in review. The data requested has been added and is satisfactory. In my opinion also the comments raised by the other reviewers have been dealt with carefully. I think this is an impressive experimental piece of work that should be published in Nature Communications.

Reviewer #3 (Remarks to the Author):

The authors have thoroughly addressed all of the comments and issues raised in the prior reviews.

AUTHORS' RESPONSE:

We would like to thank the reviewers for their careful reading and previous comments and suggestions.